# Macrophages regulate vascular smooth muscle cell function during atherosclerosis progression through IL-1β/STAT3 signaling

Yuzhou Xue[1,2,7], Minghao Luo[1,7], Xiankang Hu[1], Xiang Li[1], Jian Shen[1], Wenyan Zhu[3,4], Longxiang Huang[1], Yu Hu[1], Yongzheng Guo[1], Lin Liu[5], Lingbang Wang[6] & Suxin Luo[1✉]

Vascular smooth muscle cells (VSMCs) play a central role in atherosclerosis progression, but the functional changes in VSMCs and the associated cellular crosstalk during atherosclerosis progression remain unknown. Here we show that scRNA-seq analysis of proximal adjacent (PA) and atherosclerotic core (AC) regions of human carotid artery plaques identifies functional alterations in macrophage-like VSMCs, elucidating the main state differences between PA and AC VSMCs. And, IL-1β mediates macrophage-macrophage-like VSMC crosstalk through regulating key transcription factors involved in macrophage-like VSMCs functional alterations during atherosclerosis progression. In vitro assays reveal VSMCs trans-differentiated into a macrophage-like phenotype and then functional alterations in response to macrophage-derived stimuli. IL-1β promotes the adhesion, inflammation, and apoptosis of macrophage-like VSMCs in a STAT3 dependent manner. The current findings provide interesting insight into the macrophages-macrophage-like VSMC crosstalk, which would drive functional alterations in the latter cell type through IL-1β/STAT3 axis during athero-sclerosis progression.

[1] Department of Cardiology, the First Affiliated Hospital of Chongqing Medical University, Chongqing, China. [2] Department of Cardiology and Institute of Vascular Medicine, Peking University Third Hospital, Beijing, China. [3] Medical Department, Yidu Cloud (Beijing) Technology Co., Ltd., Beijing, China. [4] Chongqing Engineering Research Center of Pharmaceutical Sciences, Chongqing Medical and Pharmaceutical College, Chongqing, China. [5] Department of Dermatology, the First Affiliated Hospital of Chongqing Medical University, Chongqing, China. [6] Department of Orthopedic Surgery, the First Affiliated Hospital of Chongqing Medical University, Chongqing, China. [7] These authors contributed equally: Yuzhou Xue, Minghao Luo. ✉email: luosuxin0204@163.com

Atherosclerosis and its complications, including myocardial infarction, ischemic cardiomyopathy, stroke, and peripheral arterial disease are the main causes of cardiovascular disease-associated morbidity and mortality. Atherosclerosis is a chronic, lipid-driven vascular inflammatory disease involving the pathophysiological activation of multiple cell types, including immune cells, vascular smooth muscle cells (VSMCs), and endothelial cells[1,2].

VSMC phenotypic switching is often used to describe the de-differentiation of contractile VSMCs to a synthetic state. However, this term has been extended to a number of phenotypic states observed during atherosclerosis development[3,4]. The trajectories of VSMC differentiation regulate plaque growth and stability. However, the specific contribution of VSMC subsets to atherosclerosis progression is not fully understood. For example, macrophage-like VSMCs are likely contributed to plaque instability, in light of their pro-inflammatory profile[5]. However, current research indicates that macrophage-like VSMCs are functionally different from myeloid-derived macrophages[6,7].

The various roles of immune cells, especially macrophages, have been unraveled with the development of single-cell analysis technologies[1,8,9]. Three main macrophage clusters, namely: resident-like anti-inflammatory, inflammatory, and TREM2[high] macrophages, have been identified in human atherosclerotic plaques[10]. And, two more macrophage subset, including IFNIC and cavity macrophages have been revealed by a meta-analysis based on mouse data[11]. The inflammatory macrophages identified from both human and mouse data, forming the major macrophage population within the plaque intima, are enriched for numerous pro-inflammatory factors[12].

Interactions between VSMCs and macrophages have been previously demonstrated in vitro[13]. In cell culture, VSMCs produce multiple cytokines and chemokines, which can induce macrophage activation, and thus amplify the innate immune response[14]. In turn, macrophage regulates the clonality and apoptosis of VSMCs during atherosclerosis[15,16]. Overall, the interactions between macrophages and VSMCs throughout atherosclerosis progression have not been comprehensively studied through single-cell transcriptomics.

Herein, we used single-cell RNA-sequencing (scRNA-seq) to identify the different cell subsets between the proximal adjacent (PA) and atherosclerotic core (AC) portions of plaques. VSMC functional alterations and crosstalk between different subtypes of VSMCs and macrophages were also been explored. Then regulatory effect of macrophages on macrophage-like VSMCs was further examined through in vitro and in vivo tests.

## Results

**scRNA-seq revealed different cell populations during atherosclerosis progression.** A total of 35,021 cells were clustered, and 19 cell populations were identified (Fig. 1a). Cells from two atherosclerosis groups (PA vs. AC) were also properly integrated (Fig. 1b). The differently expressed markers genes of each cluster listed in Supplementary Table 1 were used for cluster annotation. Based on the differential expression of established lineage markers, 8 main cell types were assigned to 19 different clusters (Fig. 1c, e and Supplementary Table 2). Three cell clusters (clusters 3, 4, 15) highly expressed VSMC markers (*TAGLN, ATAC2, MYH11, PDGFRB*; Supplementary Fig. 1a)[17–19]. Myeloid cell markers (*CD68, CD14, CX3CR1, LYZ*) were upregulated in clusters 5, 7, 8 (Supplementary Fig. 1b)[20–22]. Furthermore, we identified 5T cell clusters (clusters 0, 1, 6, 10, and 17; expressing *CD45* and *CD3*)[23], 3 endothelial cell clusters (clusters 2, 12, and 16; expressing *CD34* and *PECAM1*)[24], 2 dendritic cell clusters (clusters 13 and 18; expressing *CD1C* and

*CD123*)[25], 1 B cell cluster (cluster 1; expressing *CD79A* and *CD79B*)[26], 1 stem cell cluster (cluster 14; expressing *KIT* and *CD44*)[27,28], and 1 NK cell cluster (cluster 11; expressing *CD161* and *NKG7*)[29]. The complex heatmap and dot plot for the selected markers are shown in Supplementary Fig. 1c, d. Heatmap of similarity to known cell type (Supplementary Fig. 2a) and intersection analysis of top markers with VSMC subset in Depuydt's data (Supplementary Fig. 2b) validate the population identities of cluster 3, 4, 15[20]. Furthermore, the number of common genes of top markers in cluster 3, 4, 5, 7, 8, 15 with different cell types in Wirka's mouse data have been compared, indicative of the consistent population identities as we defined before (Supplementary Fig. 3a–f).

A significant difference in cell type composition was observed between PA and AC groups ($P < 0.001$, Fig. 1d and Supplementary Table 3). The higher percentages of myeloid (9% vs 22%) and T (28% vs 39%) cells indicated immune activation, whereas the abundance of VSMCs (25% vs 19%) and endothelial cells (28% vs 10%) decreased during atherosclerosis progression. However, the abundance of other cell types did not differ between the PA and AC group. To further explore the alterations in cell types, we focused on VSMC and myeloid cell subsets changes due to their central role in atherosclerosis[5,30].

Extracting and re-clustering VSMCs to identify the intrinsic heterogeneity, revealed 5 clusters in both the PA and AC groups (Fig. 2a, b). We assigned 4 distinct phenotypes to different clusters based on marker expression (Fig. 2c and Supplementary Fig. 4a). The top markers for each cluster are listed in Supplementary Table 4. Cluster 0 cells, highly expressing contractility markers (*TAGLN, ACTA2, MYH11*), were identified as contractile VSMCs (Fig. 2d, e and Supplementary Fig. 4a). Cluster 1 and 4 expressing macrophage markers (*CD68, LGALS3, CXCR4, CD74, PTPRC*) were defined as macrophage-like (Fig. 2d, e and Supplementary Fig. 4a)[7,31]. Furthermore, cluster 2 and 3 were defined as fibroblast-like (expressing *PDGFRβ, ELN,* and *FN1*) and chondrocyte-like (expressing *BMP2 and RUNX2*), respectively (Fig. 2d, e and Supplementary Fig. 4a)[4,32]. There was no significant difference ($P = 0.21$) in subset composition between the PA and AC groups (Supplementary Table 5). However, pathway analysis of dysregulated genes in macrophage-like VSMCs compared with other VSMC subsets indicated contractile and pro-inflammatory functions (Fig. 2f), which suggested potential functional alterations during atherosclerosis progression. The differential expression analysis of VSMCs between the PA and AC group revealed 323 differentially expressed genes, with 27 genes for the AC group and 296 for the PA group (Fig. 2g and Supplementary Table 6). And the hallmark gene sets of gene set enrichment analysis (GSEA) were related apoptosis, myogenesis, TGF-β, and P53 pathways (Supplementary Fig. 4b), which indicates the proliferation and cell cycle involvement. Furthermore, both enrichment analysis of upregulated and downregulated genes in the AC group supported the functional alterations in VSMCs, including regulation of insulin-like growth factor (IGF) transport and uptake, IL-1B signaling pathway, and etc (Supplementary Fig. 4c, d). Hence, we further explored the relative upregulation of *CDNK1A* and downregulation of *IGF1* in the AC group indicated lowers proliferation and a prolonged cell cycle during atherosclerosis progression (Supplementary Fig. 4e, f)[33,34].

Among myeloid cells of the PA and AC groups, we identified 3 clusters assigned to the 3 canonical macrophage subsets, including resident-like, inflammatory, and *TREM2*[high] macrophages (Supplementary Fig. 5a–c). The differentially expressed genes in each cluster are listed in Supplementary Table 7. Cells of cluster 0 were recognized as *TREM2*[high] owing to their high expressing *CD9, LGALS3, CTSB,* and *ABCA1*, which are gene

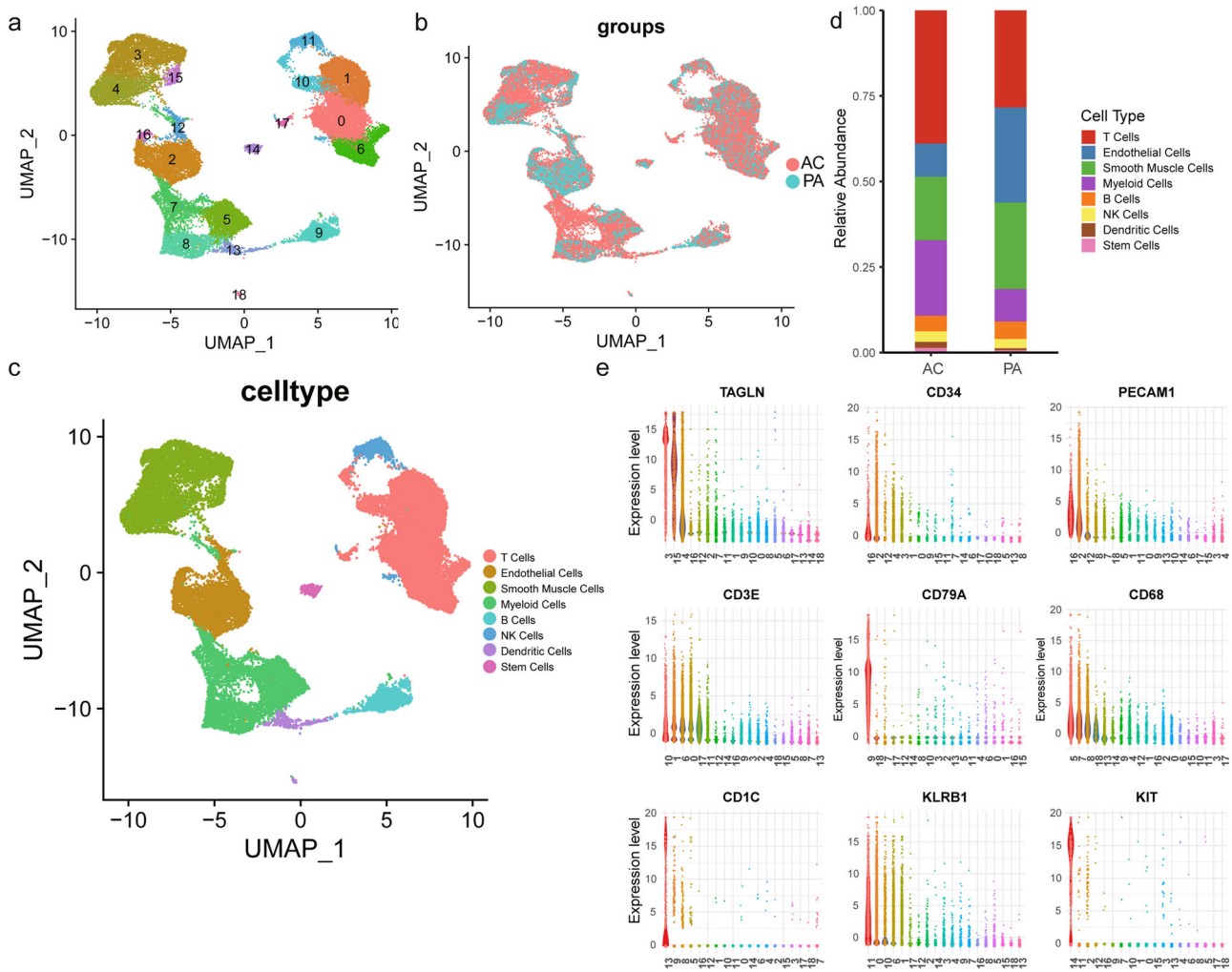

**Fig. 1 Single-cell RNA sequencing (scRNA-seq) analysis reveals cellular heterogeneity of proximal adjacent (PA) and calcified atherosclerotic core (AC) portions of plaques.** UMAP clustering plots showing **a** 19 color-coded cell clusters, **b** different groups, and **c** 8 cell populations of a total of 35,021 individual cells based on gene expression. **d** Bar chart of the relative frequency of different cell types in PA and AC groups. **e** Violin plots of signature genes of different cell type confirmed cluster identities.

markers of Trem2+ macrophage as previously defined (Supplementary Fig. 5d, e, g)[35,36]. Furthermore, cluster 3, exhibiting an upregulation of *CX3CR1* as well as anti-inflammatory markers *FOLR2* and *MRC1* was identified as a resident-like subset (Supplementary Fig. 5d, e, g)[37,38]. In cluster 2, we detected the predominant upregulation of the inflammation associated genes (*NFKBIA, IL1B, CXCL2, S100A12, S100A9, S100A12*), suggestive of an inflammatory phenotype (Supplementary Fig. 5d, e, g)[12,39]. We generated a feature plot of selected genes (Supplementary Fig. 6a). And, the intersection analysis of top 100 genes in different clusters with top markers in different macrophage cell type identified in Zernecke's meta-analysis further validates the macrophage subtype identification (Supplementary Fig. 6b–d). An increased abundance of *TREM2^{high}* (19 to 40%) and a relatively decreased abundance of resident-like (39 to 33%) macrophages were observed between the PA and AC group (Supplementary Fig. 5f and Supplementary Table 8), indicative of foamy cell accumulation and anti-inflammatory cell pruning in the AC group. Furthermore, inflammatory macrophages, forming the major population (42%) in the PA group, were associated with inflammatory activation within lesions. Importantly, their percentage decreased considerably (27%) at the late stage of atherosclerosis.

**Pseudotime trajectory revealed a key role of macrophage-like VSMCs in atherosclerosis progression.** The cells in cluster 3, 4, and 15, which are defined as VSMCs, are included in the following Monocle analysis. Pseudotime analysis ordered VSMCs in a trajectory and identified 5 different states based on gene expression (Fig. 3a, b). The root of the trajectory was mainly populated by contractile and fibroblast-like VSMCs (Fig. 3c). The second node generated two distal VSMC states, which mainly contained macrophage-like VSMCs (Fig. 3c). Furthermore, we split this trajectory based on the PA and AC groups (Fig. 3d). The distal path was mainly state 4 in the PA group and state 3 in the AC group, which indicated the state alterations of VSMCs during atherosclerosis progression. Then we found a high percentage of state 4 (35%) and low percentage of state 3 (11%) in the PA group, whereas the AC group exhibited a lower abundance of state 4 (11%) and a higher abundance of state 3 (32%) (Fig. 3e and Supplementary Table 9). Accordingly, we conducted BEAM analysis in order to identify genes associated with the second branch node. And then, the genes with branch-dependent expression were assigned them to 3 gene clusters according to trajectory differentiation to identify the mechanism by which the VSMC fate decision is made (Fig. 3e and Supplementary Table 10). Genes cluster 1 upregulated in state 4 was associated

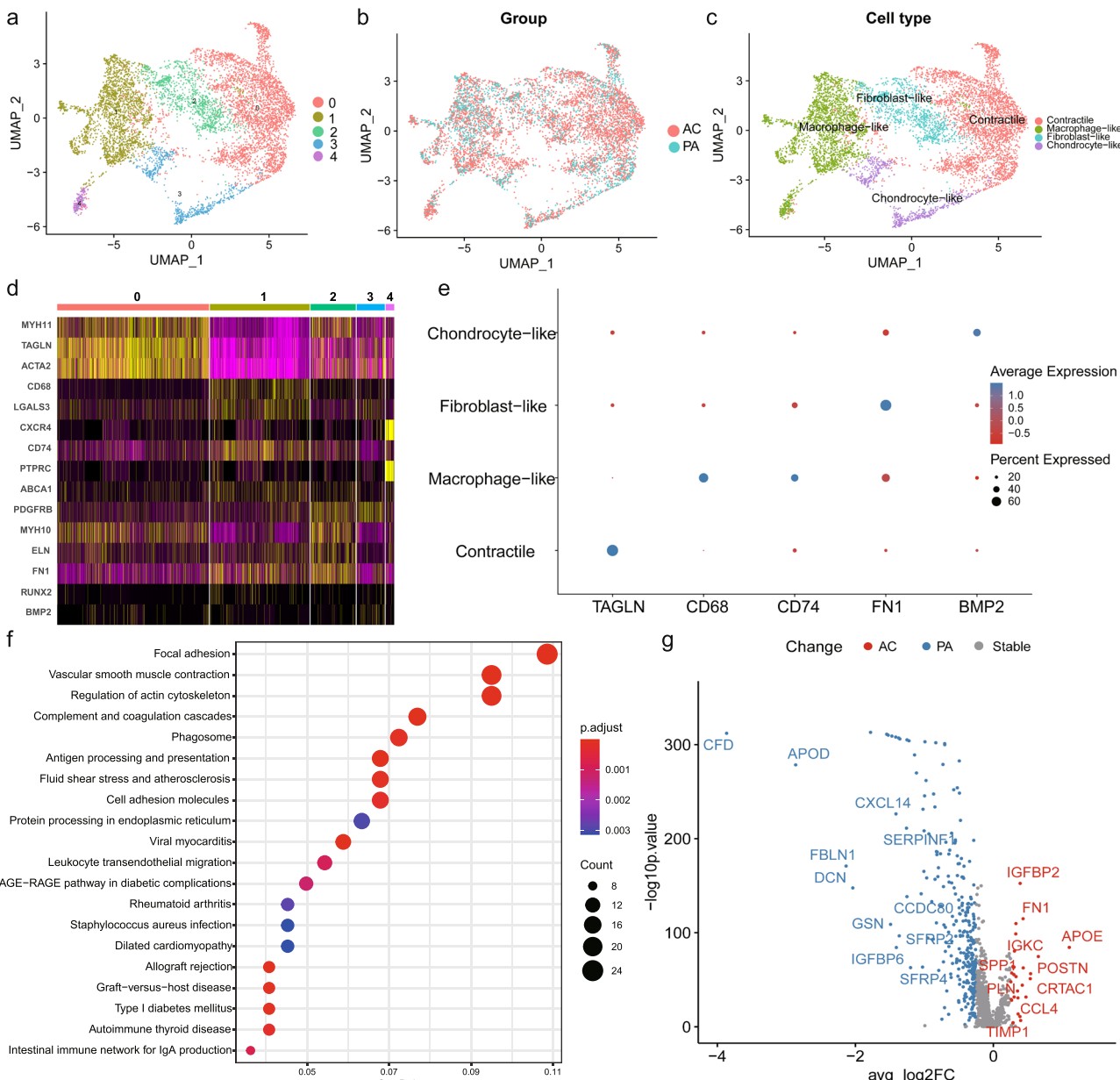

**Fig. 2 Diversity of the vascular smooth muscle cells (VSMCs) in proximal adjacent (PA) and calcified atherosclerotic core (AC) portions of plaques.** UMAP clustering plots revealing **a** 5 color-coded cell clusters, **b** different groups, and **c** 4 cell population of a total of 7053 individual cells based on gene expression. **d**. Heatmap of marker genes in different VSMC subsets per among clusters. **e** Dot plot of cluster-identifying marker genes in VSMCs. **f**. Top Kyoto Encyclopedia of Genes and Genomes (KEGG) pathways of differently expressed genes (adj.*P* value < 0.05 and |avg_log2FC| > 0.25) in macrophage-like VSMCs compared with other VSMC subsets. **g**. Volcano plot shows the differently expressed genes in VSMCs determined that were upregulated in PA or AC group (*P* value < 0.01 and |avg_log2FC| > 0.25).

with *PI3K-Akt*, focal adhesion, *MAPK*, and regulating pluripotency stem cells signals (Fig. 3f). Gene cluster 2, highly expressed in state 3 was related to cytokine-cytokine receptor interaction, chemokine signal, and viral myocarditis pathways (Fig. 3g). The top 3 upregulated and downregulated genes between PA and AC group VSMCs were selected and also exhibited differential expression trends after the second branch node (Fig. 3h). These findings indicated state alterations in VSMCs throughout atherosclerosis progression after the second branch node, which were mainly triggered via the functional modulation of macrophage-like VSMC.

**The identification of master TFs in macrophage-like VSMCs contributing to atherosclerosis progression**. In order to explore

the relationship between gene expression changes and the underlying alterations in intracellular signaling, we analyzed our data using DoRothEA, a computational algorithm for inferring TF networks. We identified aberrantly activated TFs in macrophage-like VSMCs ($n = 127$; Fig. 4a and Supplementary Table 11). Furthermore, we analyzed TFs differentially expressed between the PA and AC group. A total of 52 TFs were differentially expressed in both macrophage-like VSMCs and the AC group (Fig. 4b and Supplementary Table 12). We then analyzed activity changes for those 52 TFs among different cell states (Fig. 4c). The consistency of TF activity trends in different atherosclerotic groups (PA vs. AC) and states further indicted the central role of macrophage-like VSMCs during atherosclerosis progression. The activities of selected TFs were also plotted

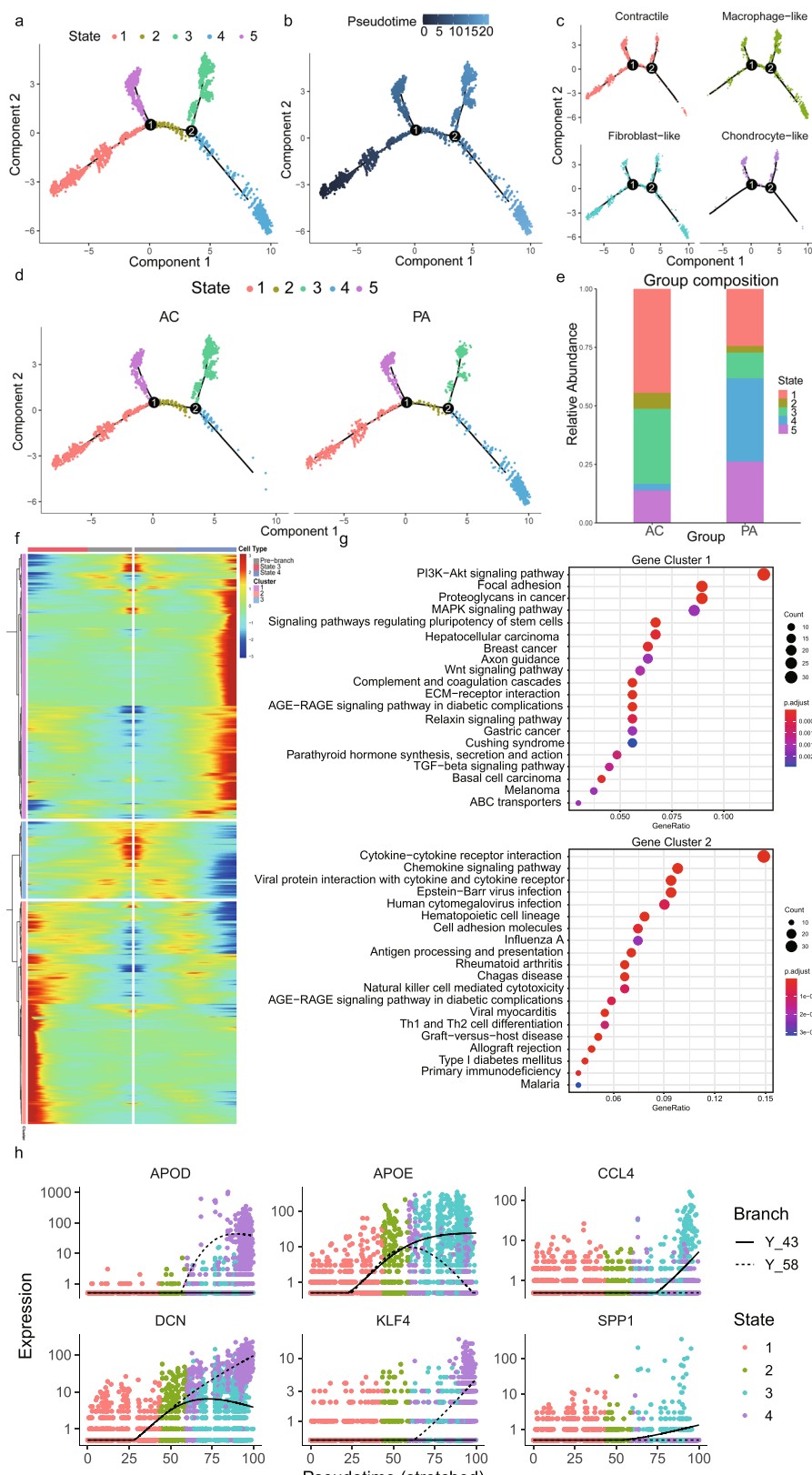

(Fig. 4d). *JUN*, involved in autophagy and cell death, was highly activated in AC and macrophage-like VSMCs[40]. Moreover, we found significant *SMAD3* and *ZEB2* regulon activation in macrophage-like VSMCs and the PA group. Both canonical TFs are associated with TGF-β activation[41]. The TF-target network for 52 TFs is showed in Fig. 4e and Supplementary Table 13.

**Alterations in macrophage-like VSMC cellular interactions**. To further understand the contribution of cellular crosstalk to macrophage-like VSMC functional alteration, we employed NicheNet. In the PA group, the cell-to-VSMC interactions were dominated by NK, myeloid, and T signals (Fig. 5a). With disease progression, we observed an increase in endothelial, myeloid and

**Fig. 3 Dynamic phenotype modulation across vascular smooth muscle cell (VSMC) spectrum in proximal adjacent (PA) and calcified atherosclerotic core (AC) portions of plaques. a** Single-cells trajectories of the 5 VSMC states through the pseudotime identified by Monocle analysis. **b** Single-cells trajectories of VSMCs colored by pseudotime. **c** Single-cells trajectories split by different VSMC subtypes. **d** Single-cells trajectories of the 5 VSMC states split by different PA and AC group. **e** Bar chart of the relative frequency of the 5 VSMC states in PA and AC groups. **f** Heatmap representing the expression dynamics of 3 gene clusters with the second branch node dependent and increased or reduced expression at State 3 and State 4 through BEAM analysis. **g** Kyoto Encyclopedia of Genes and Genomes (KEGG) terms of the upregulated gene cluster 1 and 2 in State 4 and State 3, respectively. **h** Pseudotime kinetics of indicated genes from the root of the trajectory to State 3 (solid line) and State 4 (dashed line).

smooth muscle cell signaling in AC group (Fig. 5a). Analysis of cell-to-VSMC interactions revealed several signaling pathways including IL-1β and VWF signaling. The top receptors in VSMCs including ACKR4, IL1R1, and JAM3 are involved in inflammatory processes (Fig. 5b). Hence, we focused on the interaction of macrophage-like VSMCs with other subsets of macrophages and VSMCs in the PA and AC groups (Fig. 5c, d, Supplementary Fig. 7a, and Supplementary Table 14). Inflammatory and TREM2[high] macrophages were the main contributors to functional alterations in macrophage-like VSMCs. IL-1β secreted by inflammatory macrophages was the top ligand affecting a large number of target genes in macrophage-like VSMCs (Fig. 5e and Supplementary Fig. 7b). Furthermore, we analyzed intersections between the 52 TFs and top target genes. The analysis revealed 8 master TFs (STAT3, RARG, JUND, EPAS1, ETS1, KLF13, JUN, NR3C1) associated with the crosstalk between macrophages and macrophage-like VSMCs. The schematic in Fig. 6a depicts the intracellular interactions of macrophage-like VSMCs during atherosclerosis progression. According to the dominant role of IL-1β involved in macrophage-macrophage-like VSMC crosstalk, we chose IL-1β and STAT3, one of its targets with pleiotropic effect in atherosclerosis[42–44], for the further experimental validation.

**Association of IL-1β and STAT3 expression in bulk microarray data.** To validate the regulatory relationship between ligands secreted from macrophages and target genes in the bulk RNA-seq dataset, we selected IL-1β from inflammatory macrophages and STAT3 expressed in macrophage-like VSMCs for subsequent analysis. We determined a significant negative association between IL-1β and STAT3 expression in GSE43292 ($P = 3.5 \times 10^{-4}$, $R = -0.43$; Fig. 6b). IL-1β was evidentially upregulated, while STAT3 was significantly downregulated in the atheroma plaque group (Fig. 6c, d), which indicated potential key roles for both in atherosclerosis progression. Furthermore, the immune cell type proportions were estimated by CIBERSORT based on bulk tissue gene expression profiles, and we observed the main fractions of immune cell were macrophage subsets (macrophage M0 and M2), which were significantly activated in atheroma plaque group (Supplementary Fig. 8). And, the expression of IL-1β was inversely related to the proportions of macrophage subsets (macrophage M0, M1 and M2), whereas the level of STAT3 was positively correlated with the proportions of macrophage subsets (macrophage M0 and M2; Fig. 6e). Accordingly, we propose that IL-1β secreted by inflammatory macrophage induces the functional alterations in macrophage-like VSMCs through downregulating STAT3, and this signaling cascade may play as a main driver in atherosclerosis progression.

**Trans-differentiation of VSMCs to macrophage-like VSMCs is induced by macrophage-derived stimuli.** To validate the crosstalk of macrophage-like VSMCs with macrophages and the regulatory relationship between IL-1β and STAT3, we conducted in vitro assays to explore potential mechanisms. First, we measured the levels of IL-1β in the medium of macrophages treated

with ox-LDL. IL-1β protein levels were significantly increased upon treatment with 25 μg/ml of ox-LDL, as confirmed via both western blot analysis (Fig. 7a) and Enzyme-linked immunosorbent assay (ELISA; Fig. 7b). Culturing of VSMCs with RAW264.7 medium (RM) resulted in the continuously significant downregulation of smooth muscle cell marker α-SMA and upregulation of macrophage marker CD68 at 12 h and further time points, indicative of the trans-differentiation in VSMCs toward a macrophage-like phenotype (Fig. 7c). Furthermore, we measured the levels of STAT3, p-STAT3, and EPAS1, the downstream TFs of IL-1β, in VSMCs at different time points. STAT3 levels did not change under RM treatment (Fig. 7d). The active form of STAT3, p-STAT3, reached a peak at 12 h, then strikingly decreasing at 48 h (Fig. 7d). Moreover, EPAS1 was continuously upregulated under prolonged RM treatment (Fig. 7d).

Furthermore, we also conducted the cell co-culture experiments with BMDMs. The secretion of IL-1β increased with ox-LDL stimulation in BMDMs (Supplementary Fig. 9a). Also, the expression trends of CD68 and α-SMA in VSMCs treated with MM were consistent with those treated with RM (Supplementary Fig. 9b–d). The upregulated EPAS1 and downregulated p-STAT3 further validate the relationship between IL-1β and downstream TFs (Supplementary Fig. 9e–g). These results confirmed that the expression of STAT3 and EPAS1 in macrophage-like VSMCs is regulated by IL-1β, as initially suggested by scRNA-seq data.

**IL-1β promotes macrophage-like VSMC functional alterations by suppressing STAT3 activation.** Addition of IL-1Ra blocked the IL-1β-induced downregulation of p-STAT3 and EPAS1 (Fig. 7e and Supplementary Fig. 9e–g). Furthermore, we compared functional alterations between early- and late- stage macrophage-like VSMCs, which has been treated with RM at 12 h and 48 h, respectively (Fig. 7f). Adhesion (ICAM-1, VCAM-1), inflammation (p-p65, p65, MCP-1), and apoptosis (Bax, Bcl-2, caspase-3) markers were upregulated in macrophage-like VSMCs treated with RM for 48 h, suggestive of enhanced macrophage-like VSMC functional alterations, which was reversed via IL-1Ra (Fig. 7g–i). VSMCs were treated with Colivelin, an activator of STAT3, in order to explore the role of STAT3 in macrophage-like VSMC functional alterations. Colivelin also reversed the expression of adhesion (ICAM-1, VCAM-1), inflammatory (p-p65, p65, MCP-1), and apoptotic (Bax, Bcl-2, caspase-3) factors in macrophage-like VSMC, which indicated that the IL-1β-mediated regulation of macrophage-like VSMC functional alterations may occur in a STAT3-dependent manner (Fig. 7g–i). In order to confirm that the observed functional changes were mediated via the dysregulation of IL-1β/STAT3 axis, we conducted functional rescue assays. The inhibition of IL-1β via IL-1Ra failed to reverse the functional alterations process of macrophage-like VSMCs when Stattic was used to suppress STAT3 activity (Fig. 7g–i).

Also, functional experiments were employed to validate the changes in macrophage-like VSMCs incubated with RM (Supplementary Fig. 10a). The Oil-red O staining results demonstrated the intracellular lipid droplets were significantly

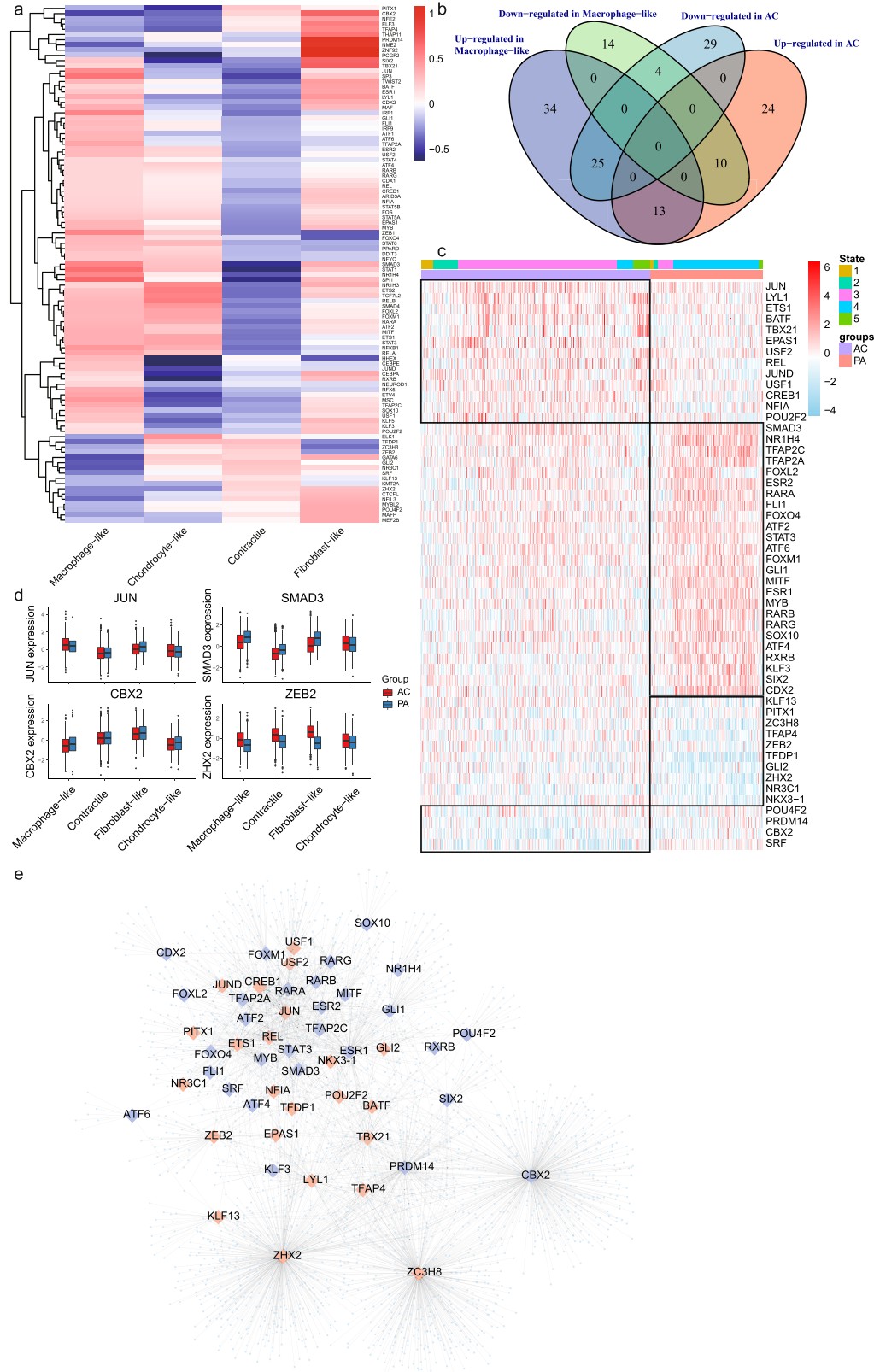

increased at 48 h and decreased in response to IL-1Ra treatment (Supplementary Fig. 10b). Consistent with declined lipid accumulation, IL-1Ra also abrogated the promotive effect of RM treatment on migration (Supplementary Fig. 10c). The outcomes further indicate proinflammatory role of IL-1β on macrophage-like VSMCs.

**IL-1β induces downregulation of STAT3 during atherosclerosis progression by in vivo tests**. The increase of IL-1β in aorta at 16 weeks of HFD feeding can be reversed by injection of AAV-sgIL-1β (Supplementary Fig. 11a, b). Compared with mice fed by HFD for 8 weeks, the lower expressions of p-STAT3 and α-SMA and higher expressions of CD68 were found in the aorta of

**Fig. 4 Transcription factors (TFs) involved in macrophage-like vascular smooth muscle cell (VSMC) differentiation. a** Heatmap of macrophage-like VSMC related TFs activities ($n = 127$) among different VSMC subtypes. **b** The macrophage-like associated and differentially expressed TFs were identified as the intersection of differential expression in macrophage-like VSMC and dysregulation in different atherosclerotic groups. **c** Heatmap of macrophage-like associated and differentially expressed TFs activities ($n = 52$). The top rows represent the transcriptome signature of macrophage-like VSMCs that refer to different state of Monocle and atherosclerotic group (PA and AC). **d** Boxplots of the 4 signature TFs (*JUN*, *SMAD3*, *CBX2*, and *ZEB2*) that upregulated or downregulated in macrophage-like VSMCs and dysregulated between proximal adjacent (PA) and atherosclerotic core (AC) group. **e** Network showing 52 selected activated (red large diamonds) and repressed (blue large diamonds) TFs in AC group and their target genes (light blue small dots).

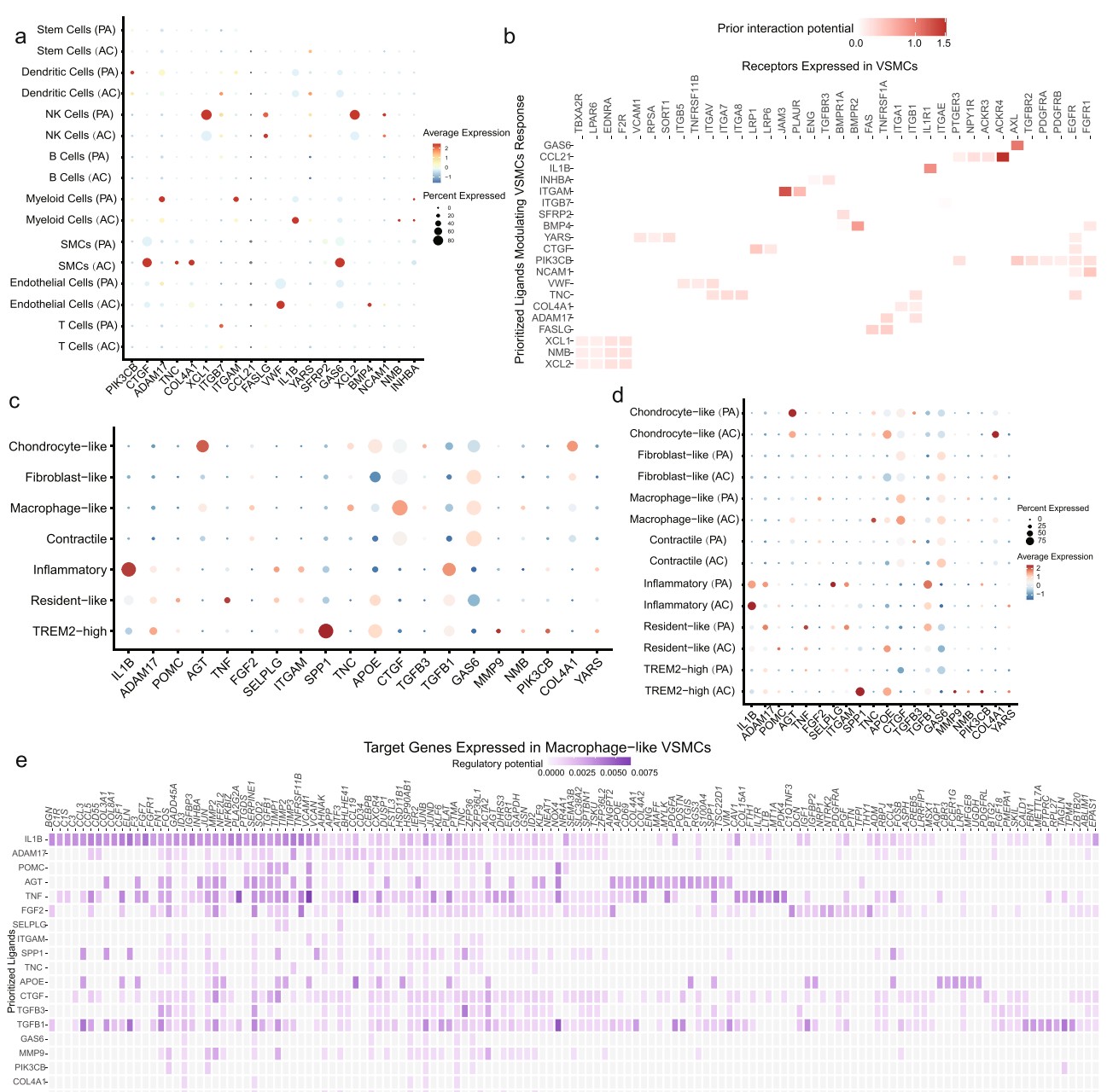

**Fig. 5 Unbiased cell–cell interaction analysis and its effect on macrophage-like vascular smooth muscle cells (VSMCs). a** Dot plots depicting which cell populations express top-ranked ligands contributing to the transcriptional response in VSMCs split by proximal adjacent (PA) and atherosclerotic core (AC) group. **b** Ligand-receptor heatmap of potential receptors expressed by VSMCs in cell crosstalk. **c** Dot plots of top-ranked ligand expressed in VSMC and myeloid cell subtypes contributing to the transcriptional response in macrophage-like VSMCs. **d** Dot plots of top-ranked ligand activities effecting contributing to the transcriptional response in macrophage-like VSMCs split by different group. **e** Ligand-target heatmap of macrophage-like VSMCs target genes of the identified Myeloid ligands.

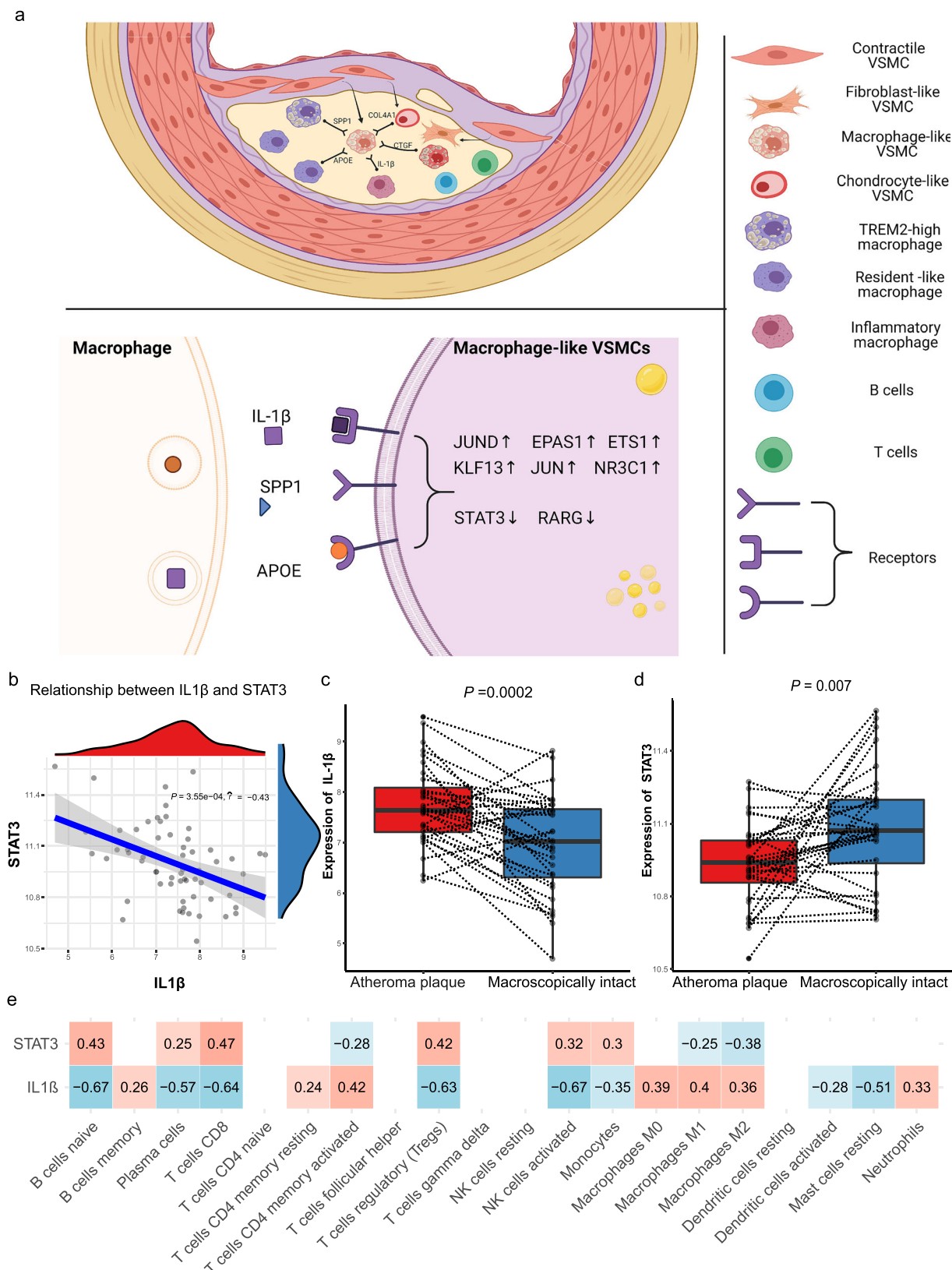

ApoE$^{-/-}$ mice fed with HFD for 16 weeks, which keep consistent with the phenomenon in cell culture experiments (Fig. 8a–d). Furthermore, upregulated of p-STAT3 expression was observed in in the aorta of ApoE$^{-/-}$ mice treated with AAV-sgIL-1β (Fig. 8d). Using immunofluorescence, expression of p-STAT3 was also repressed by IL-1β in CD68 + αSMA + cells, which were shown to

accumulate in the plaque during atherosclerosis progression (Fig. 8e–g). Furthermore, the other VSMC subtype makers have been explored among AS-8 weeks, AS-16 weeks AAV-NC, and AS-16 weeks AAV-sgIL-1β groups (Supplementary Fig. 11c–f). The upregulation of FN1, SOX9 was observed during atherosclerosis. And the higher expression of SOX9 and lower expression of FN1

**Fig. 6 The relationship between IL-1β and STAT3 in macrophage-to-macrophage-like VSMC interactions. a** Schematic illustrating the alterations of upstream top-ranked ligands and target transcription factors in macrophage-like vascular smooth muscle cells (VSMCs) in atherosclerotic core (AC) group. **b**. Linear regression plot of IL-1β with STAT3 in GSE43292 (correlation coefficient = −0.43, P value = 0.000355). Box plots show the differences in expression of IL-1β (P value = 0.002) (**c**) and STAT3 (P value = 0.007) (**d**) between macroscopically intact and atheroma plaque tissue group. **e** The correlation of IL-1β and STAT3 with the proportions of the 22 immune cell type through CIBERSORT algorithm (red means positive correlation, while blue means negative correlation). Correlation analysis was examined by Pearson test; for two groups, data were compared by paired t-test, n = 32 samples per group. The expressions were plotted along with group mean ± SE in boxplots.

were shown in AAV-sgIL-1β group, compare with AS-16 weeks AAV-NC. However, no change of CD34 expression in expression in aorta was observed among three groups. And more specific experiments should be focused on the effect of IL-1β with other VSMC subtypes.

## Discussion

In the current work, we identified functional alterations in macrophage-like VSMCs with a pivotal role during atherosclerosis progression. In the unstable plaque (atherosclerotic core), macrophage-like VSMCs exhibited enhanced apoptosis, inflammation, and adhesion. Accordingly, we defined this process as functional alterations of macrophage-like VSMCs. Furthermore, IL-1β may secreted by macrophage induced macrophage-like VSMC functional alterations through the inhibition of STAT3, which was confirmed through in vitro analyses.

VSMCs are the major cellular component of the vessel wall and key participants in both early and late-stage atherosclerosis[3,45]. VSMCs located in the medial layer of healthy artery are gradually recruited to the subendothelial space during atherosclerosis progression[46]. Further, contractile-like VSMCs are more likely to de-differentiate to other VSMC phenotypes within the atherosclerotic plaque[47,48]. In recent years, the combination of single-cell sequencing technologies with lineage-tracing methods has revealed a number of characteristic VSMC phenotypes in response to environmental stimuli during atherogenesis[20,49,50]. Accordingly, four main VSMC phenotypes were identified in our scRNA-seq data based on the marker expressions. In addition to the contractile phenotype, macrophage-like VSMCs formed the most abundant subset in our scRNA-seq data. In previous studies, in vivo and in vitro experiments revealed the trans-differentiation of VSMCs into a macrophage-like phenotype under lipid-loaden stimuli within atherosclerotic plaque[31,51,52]. However, whether macrophage-like VSMCs are detrimental to plaque stability has not yet been accurately determined[3,53]. Through differential expression analysis, we revealed that dysregulated genes between the PA and AC group were enriched in inflammatory pathways such as phagosome, antigen processed and presentation, viral myocarditis etc. Moreover, Monocle identified a VSMC fate switch from the PA to AC group, which mainly occurs in macrophage-like VSMCs (Fig. 3d). Hence, we further explored the functional and transcriptional differences of macrophage-like VSMCs between the PA and AC groups. Genes upregulated in the PA group were associated with PI3K-Akt signaling and MAPK pathway, whereas those upregulated in the AC group were more likely involved in cytokine and chemokine-cytokine receptor interactions, as well as chemokine signaling (Fig. 3g). Differentially activated TFs in macrophage-like VSMCs between the PA and AC group were also identified (Fig. 4b). Recent evidence has highlighted the critical role of macrophages during atherosclerosis progression[10,54,55]. Macrophage-like VSMCs emerge later than myeloid-derived macrophages entering plaques via the blood[49]. We investigated the cell-to-cell interactions between VSMCs and macrophages. In the subtype analysis, we observed that the effect of IL1β on macrophage-like VSMC was enhanced

(Fig. 5d). The intersection of regulatory TFs during macrophage-like VSMCs trans-differentiation process and the target genes regulated by major ligands in macrophage-to-macrophage-like VSMC interactions revealed 8 key TFs. To validate the relationship between ligands and corresponding target genes, IL-1β and STAT3 were extracted for subsequent analysis.

The clinical relevance of *IL-1β* and *STAT3* was validated in bulk RNA-seq dataset. The negative association between *IL-1β* and *STAT3* expression was also observed in plaques and paired macroscopically intact samples (*n* = 64). *IL-1β* was upregulated in plaque samples compared to corresponding macroscopically intact samples, which is consistent with the atherogenic role of IL-1β identified in several clinical trials[56,57]. Reduced expression of *STAT3* was also observed in plaque samples, but the clinical significance of *STAT3* in atherosclerosis remain unclear.

Next, we validated the potential macrophage-to-macrophage-like VSMC crosstalk via IL-1β/STAT3 axis through in vitro experiments. Since interactions between macrophages and ox-LDL are essential for inflammatory activation, we used ox-LDL to stimulate RAW 264.7 cells and BMDMs[58]. The conditioned cell medium was then used for culturing VSMCs and induced the trans-differentiation into a macrophage-like phenotype. Furthermore, IL-1β induced the macrophage-like VSMC functional alterations process (including adhesion, inflammation and apoptosis), which could be reversed via STAT3 activation. This is the first study to report the role of the IL-1β/STAT3 axis in the regulation of macrophage-like VSMCs during atherosclerosis progression. Previous research has highlighted the clinical significance of IL-1β during atherosclerosis. In 2017, the CANTOS study provided a principle of proof that canakinumab, a monoclonal antibody targeting IL-1β, significantly reduce of cardiovascular events in patients with stable atherosclerosis[56]. However, IL-1β may be the double-edged sword in atherosclerosis. Gomez and colleagues reported that IL-1β antibody treatment to Apoe$^{-/-}$ mice between 18 and 26 weeks of Western diet feeding induced a marked reduction in VSMC and collagen content and accompanied by increased lesional macrophages. Specifically, VSMC-selective *Il1r1* KO resulted in smaller lesions nearly devoid of SMC and a fibrous cap[59]. But the latest CANTOS study also indicated that canakinumab can reduce serious cardiovascular events even during long-term follow-up[57]. Furthermore, mouse experiments confirmed the efficacy of IL-1β blockade for reducing the area of established atheroma[60]. Our study also supported the detrimental role of IL-1β in atherosclerotic progression at a single-cell resolution and provided evidence for the potential mechanisms underlying the atherogenic effect of IL-1β on VSMCS.

Furthermore, recent works suggested that STAT3 is associated with the migration, proliferation, and inflammatory factor production by VSMCs[61,62]. We also observed that the down-regulation of STAT3 was related to apoptosis, inflammation, and adhesion. Interestingly, p-STAT3, the activated form of STAT3, peaked in macrophage-like VSMCs during the early stage and was then progressively downregulated. This phenomenon suggests that inflammatory macrophages may stimulate the proliferation, migration, and trans-differentiation of VSMCs at the early stage, thus promoting atherogenesis, which has been

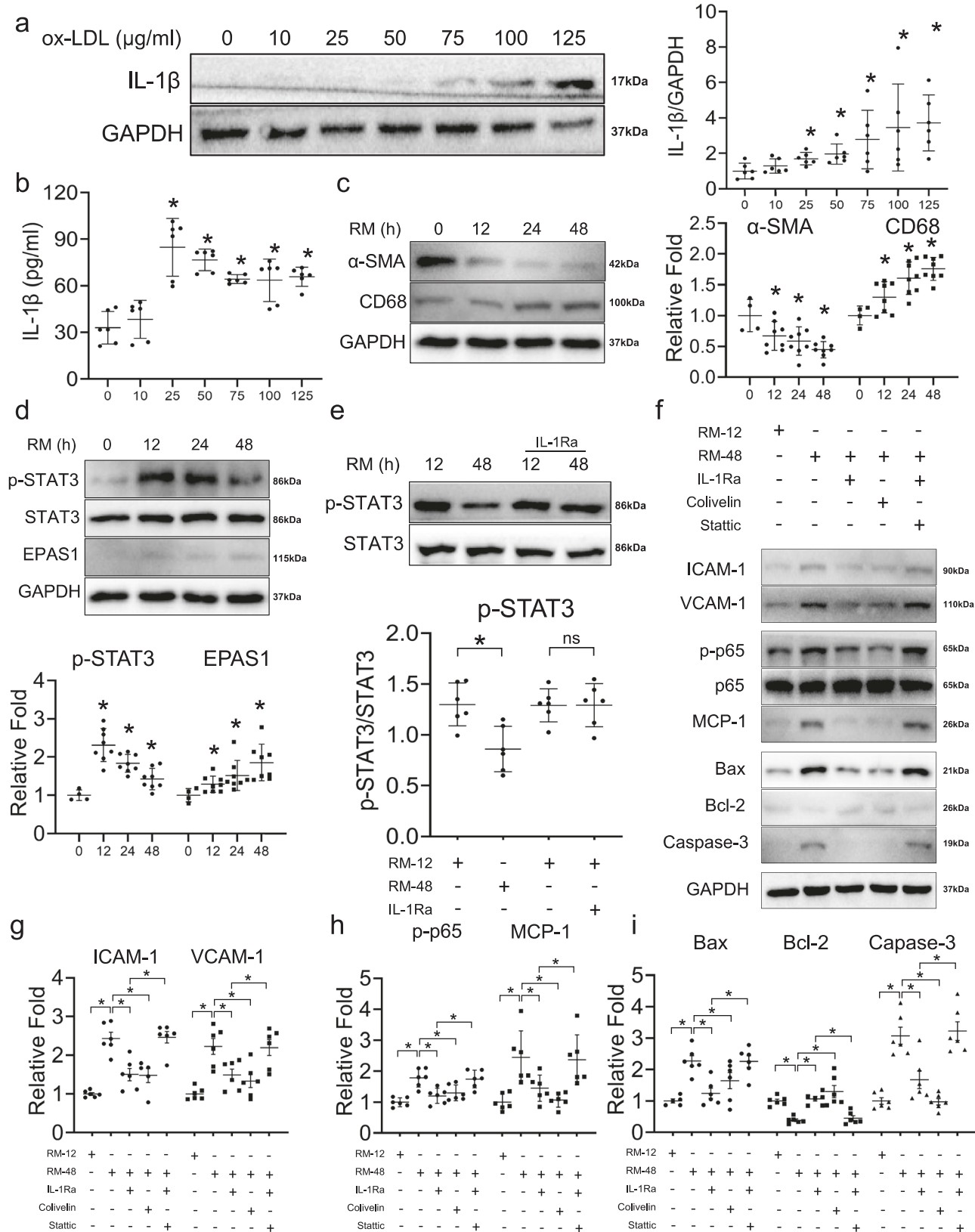

partially validated by our functional experiments. Furthermore, AAV serotype 9 was chosen to inhibit the expression of IL-1β in our animal experiments for its rapid onset, high expression, and best viral genome distribution among AAV1-9, which is similar to the effect of canakinumab[63]. The upregulated of p-STAT3 expression is also validated in the ApoE$^{-/-}$ mice through

application of AAV-sgIL-1β. Accordingly, if such macrophage-to-VSMC interactions within atherosclerotic plaque are not blocked via anti-inflammatory therapeutics, such as canakinumab, macrophage-like VSMCs may undergo enhanced adhesion, inflammation, and apoptosis thus contributing to atherosclerosis progression.

**Fig. 7 IL-1β/STAT3 axis contributing to the functional alterations process of macrophage-like vascular smooth muscle cell (VSMC). a** Western blot assay shows the expressions of IL-1β in RAW264.7 cells cultured with different concentrations of ox-LDL ($n = 6$ samples in each group). **b** Enzyme-linked immunosorbent assay (ELISA) validates the positive correlation of IL-1β and ox-LDL in RAW264.7 cells medium (RM) ($n = 6$ samples in each group). **c** The expression of macrophage marker (CD68) and VSMC marker (α-SMA) according to different durations of RM treatment shown in western blot ($n = 4$–8 samples in each group as datapoints showing). **d** The expression of downstream transcription factors (STAT3 and EPAS1) of IL-1β according to different durations of RM treatment shown in western blot ($n = 4$–8 samples in each group as datapoints showing). **e** The expression of p-STAT3 could be reversed by interleukin-1 receptor antagonist (IL-1Ra) shown in western blot ($n = 6$ samples in each group). **f** Western blot assays illustrate the effect of IL-1β/STAT3 on adhesion (ICAM-1, VCAM-1), inflammation (p-p65, p65, MCP-1), and apoptosis (Bax, Bcl-2, Caspase-3) of macrophage-like VSMC. The adhesion (**g**), inflammation (**h**), apoptosis (**i**) of macrophage-like VSMC can be reversed by inhibition of IL-1β and activation of STAT3, and the effect of inhibition of IL-1β can be reversed by inhibition of STAT3 (Stattic: the inhibitor of STAT3; Colivelin: the activator of STAT3) ($n = 6$ samples in each group). The expressions were plotted along with group mean ± SE. Significance is indicated as follows: *$P < 0.05$. For two groups, data were compared by Mann–Whitney test.

However, few limitations in our study should be declared. Firstly, although we had proved the regulatory relationship of TFs (including EPAS1 and STAT3) with IL-1β during inflammatory macrophage-to-macrophage-like VSMC interactions through in vitro analyses, the other relationship between ligands and target TFs, cell interactions with macrophage-like VSMC should be further validated in the future work. Secondly, recent literatures show that KLF4 induces the differentiation of contractile VSMC to mesenchymal, macrophage, fibroblast, and chondrocyte phenotype and then decreases the stability of atherosclerotic plaque[52,64]. However, the KLF4 expression trends of different VSMC phenotypes through atherosclerosis progression and potential role of KLF4 at late stage of macrophage-like VSMC are shortage of evidence from in vivo tests. Thirdly, myeloid specific IL-1β knockout by CRISPR-Cas9 in cells or Cre-loxP technology in animals or VSMC/macrophage reporter experiments could provide conclusive evidences in the effect of macrophage-derived IL-1β on STAT3 in macrophage-like VSMCs. But it's unable to conduct these experiments in our study due to restrictions of laboratory condition. Furthermore, the inverse association of STAT3 with IL-1β in macrophage-macrophage-like VSMC crosstalk would be better validated in primary human artery smooth muscle cells (HASMCs).

In summary, our study revealed the macrophage-like VSMC functional alterations during atherosclerosis progression. The functional alterations of macrophage-like VSMCs in plaques compromises plaque stability. We identified interactions between activated macrophages and macrophage-like VSMCs, which may play a critical role in atherosclerosis progression. Finally, the role of the IL-1β/STAT3 axis in macrophage-like VSMC functional alterations was also unraveled through in vitro experiments.

## Methods

**Data collection**. Single-cell transcriptomic data (10x Genomics) of AC and patient-matched PA portions of plaques from 3 patients undergoing carotid endarterectomy (GSE159677) were obtained from the Gene Expression Omnibus (GEO) database[65]. The microarray gene expression profile (GSE43292) and corresponding plaque severity classification information were also downloaded from the GEO database for subsequent bulk RNA analysis[66].

**scRNA-seq analysis**. Gene-cell count matrices were imported in R 3.6.3 and analyzed using the Seurat package[67]. Cells expressing <500 or >5000 genes were excluded as non-cells or cell aggregates. Cells with a percentage of mitochondrial or ribosomal genes >10% or 5%, respectively, were also filtered out. Finally, a total of 35021 individual cells were included in subsequent analysis. The UMI count data was normalized and scaled via SCTransform method, which preserves sharper biological distinction compared to the log-normalized workflow[68]. After integrating all samples based on the top 5000 variable genes, clusters were generated with the first 50 principal components through principle component analysis. Clustering tree is useful for visualizing the relationship between clusters at multiple resolutions. The resolutions were determined through the clustering tree once an uns cluster observed in our analysis[69]. Cells clusters were visualized via a Uniform Manifold Approximation and Projection (UMAP) plot[68]. Different clusters were annotated using a combination of the singleR package and manual interpretation[70]. Then, intersection analysis and hypergeometric tests between top markers of

different clusters in our study with published data were used to validate the cluster identities. Expression of marker genes was plotted using Seurat functions DoHeatmap and DotPlot.

VSMCs and macrophages identified as VSMC and myeloid clusters in the integrated UMAP were then extracted and re-clustered to identify different cell subsets. After separation and grouping with "subset" command, the above procedure was applied in the second round sub-clustering analysis[71]. For differential expression analysis between subsets, we use the FindMarkers function in Seurat in order to generate average logFC and *P*-value. Gene ontology (GO) and Kyoto Encyclopedia of Genes and Genomes (KEGG) analyses were performed using the clusterProfiler package[72].

Pseudotime trajectory analysis was conducted using the R package monocle 2.18[73]. Pseudotime ordering was analyzed using the DDRTree algorithm with max components set at 2. Cells expressing traditional VSMC markers such as ACTA2 and TAGLN were set as the original point of pseudotime. Branch point analysis was performed using the BEAM function. Genes significantly associated with the second branch point ($q$ value $< 10^{-4}$) were plotted using the plot_genes_branched_heatmap function (num_clusters = 3). Gene clusters upregulated in different cell states were functionally annotated. The expression levels of selected genes across pseudotime in different trajectory paths were plotted via the plot_genes_branched_pseudotime function.

DoRothEA is a gene set resource including signed transcription factor (TF)-target interaction[74]. The TF-target interactions assigned into A to D confidence levels were extracted for subsequent analysis. VIPER in combination with DoRothEA was used to estimate TF activities in VSMC subsets from our scRNA-seq data. This method has consistently outperformed other tools in different simulated and real scRNA-seq datasets[75]. Furthermore, TFs that were differentially expressed between the different atherosclerotic groups (PA vs. AC) and dysregulated in macrophage-like VSMCs were identified, and the corresponding transcriptional network was visualized in Cytoscape 3.8.1.

We then used NicheNet analysis to predict the interactions between ligands secreted by macrophage and targets in VSMC subsets. NicheNet is designed to compute how the gene expression profile of a cell is influenced by intracellular interactions through combining expression data with prior knowledge on signaling and gene regulatory networks[76]. Overall, we identified VSMCs as the receiver cells. Ligand-target interactions among VSMCs and other cell types were analyzed in both atherosclerosis groups (PA vs. AC). Furthermore, we assigned macrophage-like VSMCs as "receiver cells" and other subsets of VSMCs and macrophage as "sender cells". The ligand-target interactions of macrophage-like VSMCs were then explored. Lastly, key TFs in macrophage-like VSMCs were identified based on the intersection of the top 200 target genes and dysregulated TFs in the atherosclerosis groups (PA vs. AC).

**Bulk RNA-seq analysis**. After identifying the regulatory relationship between IL-1β and STAT3, we validated their association based on bulk RNA-seq data. A total of 64 samples including the core and shoulder of plaques (stage IV and over) as well as paired macroscopically intact tissue (stage I and II) from GSE43292 were used in our analysis. The CEL files were downloaded and preprocessed using the affy package. The correlation between *IL-1β* and *STAT3* expression was determined via the Pearson test. Differences in *IL-1β* and *STAT3* expression between the atheroma plaque and macroscopically intact tissue were evaluated via the paired *t*-test. The CIBERSORT algorithm employs the LM22 gene sets to characterize the immune cell composition of complex tissue based on the gene expression data[77]. Hence, we also conducted this algorithm with default parameters to explore the associations of *IL-1β* and *STAT3* with different immune cell types. The results were plotted using the ggplot2 package.

**Antibodies**. Cell culture materials were purchased from Invitrogen (Carlsbad, CA, USA). Primary antibodies against p-STAT3 (Tyr705) (#9145, 1:1000), STAT3 (#30835, 1:2000), p-p65(Ser536) (#3033, 1:1000), p65 (#8242, 1:2000), and α-SMA (#19245, 1:2000) were purchased from Cell Signaling Technology (Beverly, MA, USA). Primary antibodies against IL-1β (sc-12742, 1:2000), CD34 (sc-7324,

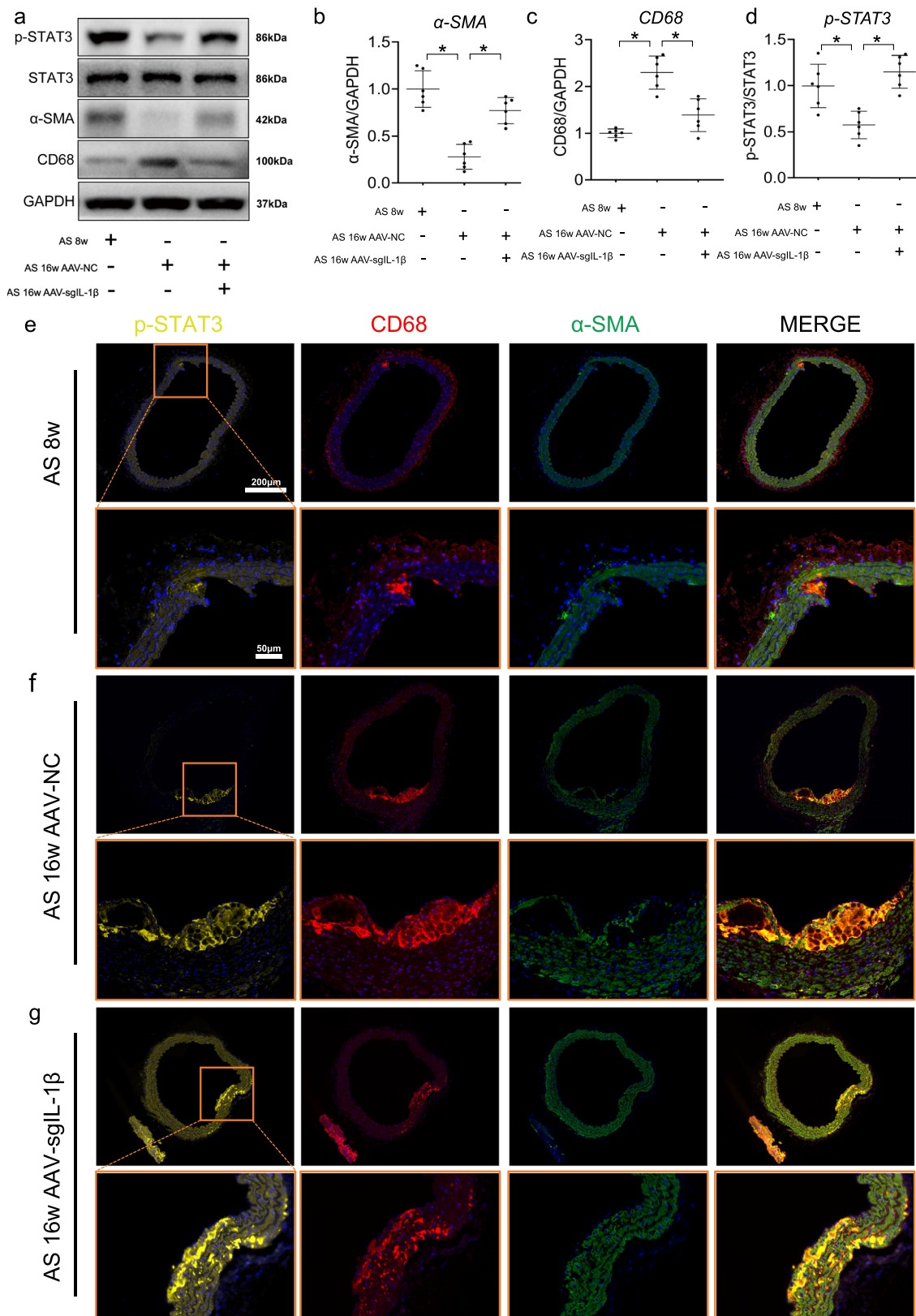

**Fig. 8 In vivo experiments identify the inverse correlations of IL-1β with STAT3 in aorta of mice. a** Western blot assays illustrate the effect of adeno-associated virus-sgIL-1β (AAV-sgIL-1β) on the expression of main SMC subsets transcription factors in vivo. Quantifications of **b** p-STAT3, **c** vascular smooth muscle cell (VSMC) marker (α-SMA), **d** macrophage marker (CD68) in aorta of mice from each group. Representative immunofluorescence staining of p-STAT3 (yellow) expressing plaque macrophage-like VSMCs in ApoE$^{-/-}$ mice at **e** 8 weeks, **f** 16 weeks, and treated with (**g**) AAV-sgIL-1β fed HFD. Eighteen ApoE$^{-/-}$ mice were randomly grouped into three groups, including high-fat diet (HFD) treated 8 weeks ($n = 6$), 16 weeks ($n = 6$), and AAV-sgIL-1β ($n = 6$) mice. Data are presented as mean ± SE. Mann–Whitney test: *$P < 0.05$.

1:2000), VCAM-1 (sc-13160, 1:2000), and EPAS-1 (sc-13596, 1:2000) were purchased from Santa Cruz Biotechnology (Santa Cruz, CA, USA). Primary antibodies against CD68 (28058-1-AP), ICAM-1 (60299-1-Ig, 1:2000), MCP-1 (66272-1-Ig, 1:2000), Bax (50599-2-Ig, 1:2000), Bcl-2 (12789-1-AP, 1:2000), cleaved-caspase-3 (19677-1-AP, 1:2000), GAPDH (10494-1-AP, 1:2000), FN1 (15613-1-AP, 1:2000), and SOX9 (67439-1-Ig, 1:2000) as well as secondary antibodies (Goat anti-mouse, SA00001-1; Goat anti-rabbit, SA00001-2) were purchased from Proteintech Group (Chicago, IL, USA). Recombinant human interleukin-1 receptor antagonist (IL-1Ra) (SRP3084) was obtained from Sigma (St. Louis, MO, USA), Stattic (inhibitor of STAT3) and Colivelin, (activator of STAT3) were purchased from MedChemExpress (Shanghai, China). Unless otherwise indicated, the remaining reagents used in this study were obtained from Sigma.

**Atherosclerotic mice.** Male apolipoprotein E–deficient (ApoE$^{-/-}$) mice on C57BL/6J back ground aged 8 weeks were fed a high-fat diet (HFD) only for 8 or 16 weeks respectively to simulate early and late atherosclerotic plaques. At the end point, mice were anesthetized with ketamine/xylazine, blood, aorta, and carotid artery were harvested after PBS perfusion. All animal experiments were approved by the local Ethics committee (The First Affiliated Hospital of Chongqing Medical University; License Number: 2021-604) and conducted in accordance with the institutional and national guidelines. To inhibit the expression of IL-1β, the mice were fixed to a stereo-locater and injected through the tail vein with 50 µl of adeno-associated virus serotype 9-IL-1β (AAV-sgIL-1β) (CMV-NLS-SaCas9-NLS-3xHA-bGHpA-U6-sgRNA, Genechem Co., Ltd., Shanghai, China) and negative control (scramble sgRNA, AAV-NC) at 8$^{th}$ week of HFD. The virus was diluted at a final titer of (1E + 11 v.g./ml) before injection.

**Cell isolations and culture.** Wild-type male C57BL/6J mice were purchased from the Laboratory Animal Center of Chongqing Medical University, and were anesthetized and killed by 1% pentobarbital sodium (1 ml/100 g) with intraperitoneal injection. Primary bone marrow derived macrophages (BMDM) were obtained by harvesting marrow from femurs of eight weeks old male C57BL/6J mice, lysing red blood cells with ammonium-chloride-potassium lysing buffer, and then culturing cells for seven days in DMEM (Gibco, Logan, Utah, USA) supplemented with 10% fetal bovine serum (FBS), 1% penicillin/streptomycin, and macrophage-colony stimulating factor (M-CSF, 20 ng/mL). All cells were regularly tested negative for mycoplasma contamination. Primary VSMCs were isolated from the aorta of wild-type male C57BL/6 J mice, using a standard tissue explants method and cultured in DMEM with 10% fetal bovine serum (Thermo Fisher Scientific, Waltham Mass, USA)[78]. Cell purity was confirmed based on α-actin immunohistochemistry and "hill-and-valley" growth pattern. VSMCs of the 5–8th generation and 90% purity were used for subsequent experiments. RAW264.7 (mouse macrophage cell line) cells were cultured in DMEM supplemented with 10% fetal bovine serum.

For in vitro atherosclerosis progression simulation and for studying the crosstalk between macrophage-like VSMCs and inflammatory macrophages, RAW264.7 or BMDM were pretreated with 50 µg/mL ox-LDL (Yiyuan Biotechnology, Guangzhou, China) for 24 h, and the medium (ox-LDL-treated RAW264.7 medium, RM or ox-LDL-treated BMDM medium, MM) was collected to culture VSMCs[79]. Early and late stage macrophage-like VSMCs were defined as co-culturing with RM/MM for 12 h and 48 h, respectively. IL-1Ra (100 ng/mL), Stattic (10 µmol/L), and Colivelin (100 nmol/L) were then added to macrophage-like VSMCs (co-cultured with RM or MM for 12 h) in order to study functional alterations in cells.

**Immunofluorescence staining.** Mice arteries were harvested, frozen in Tissue-Tek OCT media, and sliced sequentially into 10 µm sections using a cryostat. Sections were permeabilized with 0.1% Triton X-100 in PBS for 20 min and then blocked with 5% donkey serum for 1 h and incubated with related antibodies overnight at 4 °C. Next, sections were incubated for 5 min with 4′,6-diamidino-2-phenylindole (DAPI) and then for 2 h with fluorescence-conjugated secondary antibodies in a dark room at room temperature. Slices were visualized under a fluorescence microscope (Leica Microsystems, Germany).

**Western blotting.** Aortas was separated and immersed in liquid nitrogen, and then immediately transferred to an 80 °C refrigerator for storage for western blot test. Cells and tissue were lysed in lysis buffer for 60 min, followed by centrifugation at 12,000×g for 20 min. The protein concentration of the supernatant was determined via the Bradford method and 30 µg of total protein was used for western blotting. Proteins were separated via SDS-PAGE, transferred to PVDF membranes, and probed with the appropriate primary antibodies. Membrane-bound primary antibodies were detected using secondary antibodies conjugated to horseradish peroxidase. Bound proteins were finally detected via chemiluminescence (Beyotime, Shanghai, China). The gray value of bands was analyzed using Image Lab 6.0, and the level of target protein was presented as the ratio of the target protein gray value to that of the internal control GAPDH. The level of phosphorylation was indicated by the ratio of the gray value of phosphorylated protein to that of total protein.

**Enzyme-linked immunosorbent assay.** The IL-1β concentration in the culture medium of ox-LDL-treated RAW264.7 cells/primary macrophage was determined using Mouse *IL-1β* ELISA kits (Beyotime, Shanghai, China) as per the manufacturer's instructions.

**Oil-red O staining.** VSMCs were fixed in 4% PFA in phosphate-buffered saline (PBS) and then rinsed with PBS three times. The cells were then stained with Oil-red (Solarbio, Beijing) for 30 min at room temperature. Staining was assessed by optical microscopy.

**Cell migration assay.** The cell migration was assessed using the wound healing assay. Briefly, VSMCs were incubated for 24 h. Six-well plates were seeded with VSMCs and allowed to form a confluent cell monolayer. We used a 1 mL pipette tip to scratch the monolayer cells from the middle of the well followed by washes to remove the non-adherent cells. The cells were photographed at 0 h. The cells were photographed again to estimate the cell migration after 24 h.

**Reporting summary.** Further information on research design is available in the Nature Portfolio Reporting Summary linked to this article.

## Data availability
The publicly available datasets (GSE159677 and GSE43292) analyzed in this study can be found at: https://www.ncbi.nlm.nih.gov/geo/. The supplementary tables and original data underlying plots in figures are available in Supplementary Data 1 and 2, respectively.

## Code availability
In silico analysis was performed using custom R scripts designed for this study (R software 3.6.1 or 4.0.3). R scripts are available once from the corresponding author on reasonable request.

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

## Acknowledgements

We thank Dr. Jianming Zeng (University of Macau), and all the members of his bioinformatics team, biotrainee, for generously sharing their server. We also thank Helix Medical Institute (HMI, shanghai), Weifeng Hong (Fudan University), Ping Zheng (Shanghai Jiaotong University), and Hongsheng Cheng (Huazhong University of Science and Technology) for providing technology support. We thank Dr. Ali Toramani (Scripps Research Institute) and his group for generating and publicly sharing the single-cell dataset utilized in this study. This work was supported by the National Natural Science Foundation of China (82070238), the General Project of Chongqing Natural Science Foundation (cstc2020jcyj-msxmX1091), and the Intelligent Medicine Training Program of Chongqing Medical University (ZHYX202017).

## Author contributions

Y.X. and S.L. designed and supervised the project. M.L., Y.H., and Y.G. performed the experiments and collected the data. Y.X., L.L., and L.W. conducted the data analysis. X.H., X.L., W.Z., J.S., and L.H. interpreted the data, contributed to the writing of the manuscript, discussed the results and implications, and edited the manuscript. Y.X. and M.L. finally revised the manuscript.

## Competing interests

The authors declare no competing interests.
