## [Peer Review File · Communications Biology]

Reviewers' comments:

Reviewer #1 (Remarks to the Author):

The authors studied VSMC phenotypes and intracellular communication during atherosclerosis progression, by using SCS to compare proximal adjacent and atherosclerotic core regions of human plaques. They identify 8 cell types, amongst which are myeloid cells and VSMCs. Macrophage-like VSMCs seem functionally different between PA and AC according to pseudotime analysis. Transcription factors that contribute to these functional alterations were identified in silico. Moreover, intercellular crosstalk between macrophages and macrophage-like VSMCs was identified.

Data supports most of the conclusions, and the authors address a question interesting to the field: how/why do these new SMC subsets form. A great image in figure 6A helps the reader decipher the proposed communication.

Nevertheless some major concerns remain, and many details remain to be resolved.

Major concerns

- The authors have used a dataset in a public repository, generated by Torkmani et al. and only published as a preprint. Decoding the transcriptome of atherosclerotic plaque at single-cell resolution | bioRxiv . please credit these authors work appropriately.
- Annotation: to maintain consistency in the field, it is advisable to compare cell type and subset annotation and markers to previous single cell papers by Wirka (<https://doi.org/10.1038/s41591-019-0512-5>) Fernandez and Pan (<https://doi.org/10.1161/CIRCULATIONAHA.120.048378>). Specifically, Re terminology of SMCs, modified SMCs, fibroblasts, and macrophage subsets.
- Please rename the major cluster SMCs in 1D, as mesenchymal cells, as this likely also contains fibroblasts and pericytes, as annotated by Wirka. Indeed table 1 shows RGS5 in cluster 3, and DCN/Lum in cluster 4 DEGs. Also, the authors state that in figure 1 clusters express Tagln/Myh11/ACTA2 and annotate them all as VSMCs. These are then reclustered and find fibs? And then in figure 2, they show that, only in cluster 0 the Myh11/Tagln/ACTA2 expression is high, and in the other clusters (1-2-3-4) it is not? But then they proceed to analyze them as "VSMCs"
- W.r.t. sub clustering and pseudotime analysis, please indicate how proliferation and cell cycle were taken into account and show feature plots of major proliferative markers ki67/pcna cell cycle genes or violin plots of signature scores thereof.
- Please discuss the evidence for the proposed trajectories in relation to absence of reporter mice. i.e., Wirka analysed enrichment of myh11+ transcriptional profiles in human modified SMCs. This could be done by hypergeometric testing of DEGs in subsets compared to Wirka DEGs.
- How do the authors explain rise of KLF4 TF only in state 4 and 5, not 3, as it was previously identified to act in mf-like SMC differentiation? Shankman et al showed that KLF4 Ko led to less mf-like SMCs (10.1038/nm.3866)
- All human data is based on RNA level, partly derived from digested tissue, and hence, please provide validation on protein level and/or in intact carotid artery segments using immunohistochemistry and/or western blotting for main SMC subsets markers/transcription factors, and the inverse correlations of Il1b and STAT3.
- Please repeat main experiments with primary mouse macrophages such as bone-marrow derived or naïve peritoneal macrophages, or human PBMCs. Please note, thioglycolate -induced peritoneal macrophages are not naïve/resident like.
- The statement in line 386 that EPAS1 and P-STAT3 are directly related to Il1b cannot be justified by the data of figure 6. Please adjust statement. For P-stats3 this is shown in figure 7. oxLDL initiates many affects, a.o. reactive oxygen species, also able to stabilize EPAS1 levels. Please silence or block IL1b in the presence of oxLDL to support the direct relationship.
- Please avoid the term "ageing" to refer to the transition between early and late stages in the differentiation process of Mf-like SMCS.

Minor

Abstract:

- Line 35: intracellular (signalling, within cell) or intercellular (crosstalk)?

- line 37 please include source and artery
- In intro/abstract, the authors focus on VSMCs and their associated cellular crosstalk very much. However eventually, they focus more on the effect of other cells (macrophages) on the VSMCs.

Introduction

- Writing not always great (line 111: doesn't flow well, line 112: starting sentence with "But", Line 114 and 117: comma missing, line 122)
- A little unclear whether focus on macrophage-like VSMCs, or the crosstalk between macrophages and VSMCs, or both? Hypothesis? Will the macrophage-like VSMC interact with macrophages, or what do you expect?
- line115-117: please include "have been identified in human atherosclerotic plaques". And include types described in mice (Zernecke et al, meta-analysis.)

Methods

- Several textual mistakes/unclarities: line 134, line 143, line 148, 174, 225, 227, 242
- Very detailed (scRNA-seq analysis), methodology seems correct and applicable
- Line 209: C57BL/6N or C57BL/6J?
- Explain why only male mice were used for in vitro experiments. Why not use human cells? For further validation of in silico data in human situation.
- Gender of human patients not stated
- Line 136-8: what is meant by "The microarray gene expression profile 137 (GSE43292) and corresponding Stary classification of plaques were also downloaded from the 138 GEO dataset" is this the same patients or others.
- Line 236; how was oxLDL generated or what was its source?

Results

- Results difficult to follow at times
- Figure 1E: Add individual datapoints, and number of cells/group.
- Figure 2F does this represent DEGs across all 4 subsets? It seems more insightful to also show DEGs specific for each subset?
- Several textual mistakes/unclarities: line 252, 313, 362, 363
- Line 251: state more clearly that info can be found in supplements (Table 1, but applies to whole manuscript)
- Line 288-292: you speak of cluster 0, 2 and 3, whereas in the figure, you speak of cluster 0, 1 and 2
◇ confusing, please change
- Figure 4A, please also provide this list as a table to allow colleagues to re-use data, as font in 4A is very small.
- In general: switch PA and AC results around (PA first in graphs, then AC)
- Line 331-332: The consistency of TF activity trends in different atherosclerotic groups (PA vs. AC) and states further confirmed the central role of macrophage-like VSMCs during atherosclerosis progression. ◇ based on in silico analysis only. Further validation is required.
- Figure 5: in text: the top receptors in VSMCs including ACKR4, IL1R1 and JAM3 are involved in inflammatory processes. Hence, we focused on the interaction of macrophage-like VSMCs with other subsets of macrophages and VSMCs in the PA and AC groups. ◇ Did you also analyze ligand-receptors of the macrophage-like VSMCs specifically?
- Authors assessed ligands that contribute to the transcriptional response in macrophage-like VSMCs. Vice versa, do these macrophage-like VSMCs also affect the transcriptional response of other plaque cells?
- Authors should more clearly explain why STAT3 and IL1b are chosen to validate further. Is STAT3 a target of IL1b?
- Figure 5D font size is not readable
- Line 360-2 figure 8 should be figure 6,
- Please introduce in the results how the immune cell fractions were (acquiRed i.e., CIBERSort)
- Line 363: why subdivision into M0/M1/M2 macrophages here, whereas TREM2hi/pro-inflammatory/resident earlier?
- Figure 6E placement is confusing, possibly flip orientation and place below B-D. in the legends

explain color-coding of red and blue, and what it means if no coefficient is shown..

- 6 The figure is unclear ◊ IL1b is inversely related to the proportions of mf subsets, and STAT3 positively correlated with proportions of mf subsets. ◊ isn't IL1B positively correlated, and STAT3 inversely?
- Line 367-370: "Accordingly, we propose that IL-1B secreted by inflammatory macrophages induces the functional alterations in macrophage-like VSMCs through downregulating STAT3, and this signaling cascade is the main driver of atherosclerosis progression." ◊ this is only based on in silico analysis and correlations. So please adjust the statement to avoid overinterpretations
- Figure 7: why use mouse macrophages? Why macrophage-like VSMC "aging"?
- Line 378; explain "RM" here too
- Line 389: what is the function of IL-Ra, static, and Colivelin? This is not mentioned in methods. In the methods its name is IL-1Ra
- Line 391: how did the authors acquire early and late STAGE Mf-like VSMCs?
- Figure 7 please indicate number of biological (mice) and technical replicates (wells per assay condition), as well as number times the expt was repeated in the legend. Enlarge size of scatter in plot: signs are too small to see replicates. Please adjust layout of figure 7 to avoid confusion: swop quantification of 7A with 7C; o second row show 7B, 7C western and 7c quantifications. Indicate statistics for multiple groups shown in figure 7, and confirm normal distribution of data to permit T-test instead of non-parametric

Reviewer #2 (Remarks to the Author):

In the paper by Xue et. al., the authors use single cell gene expression analysis of human atherosclerotic plaque or adjacent proximal regions to better predict potential regulatory interactions between key cell types in disease progression. They use this data to describe clusters associated with different regions of the aorta to focus primarily on smooth muscle cell diversity. Furthermore, they use prediction approaches to propose potential interactions between essential cell types known to function in atheroma formation, specifically macrophage and smooth muscle cell. Their final conclusions are that macrophages regulate smooth muscle function in atherosclerosis, but data supporting this conclusion is limited beyond predictive algorithms.

The study design is innovative and of potential interest to the field. However, a number of major concerns exist regarding data analysis, presentation, and interpretation. The primary concern is the over interpretation of results, in particular drawing conclusions from informatics approaches without showing sufficient follow-up experiments. And second, potential contradictions in the data presentation that make it difficult to determine what is being presented and whether the conclusions being presented are relevant to disease. A number of specific requests are outlined below that may improve the clarity of the paper and make it more accessible to potential readers.

Major Concerns:

1. CD68 expression appears to be absent in smooth muscle cell clusters in Figure I and Supplemental Fig I, but in subsequent figures CD68 expression is a key defining marker for a smooth muscle cell population of interest. This needs further explanation to understand where the smooth muscle cell cluster is deriving from.
2. It was surprising to me that clusters (particularly macrophage) representation didn't appear to be dramatically different between the sources of cells isolated. Could the authors explain this observation, since it is contradictory to what one might typically expect to find.
3. The authors use "macrophage-like" in a number of contexts to describe a smooth muscle cell cluster type. While this nomenclature has been used sporadically in previous publications, there is little data provided beyond CD68 gene expression to suggest similarity between these cells and true macrophages. Perhaps the cells may be more accurately described as CD68+ smooth muscle cells- since this is all that is shown in the manuscript. If the authors performed more in depth comparative analysis or even functional experiments, it may become reasonable to call the cluster "macrophage-

like”.

4. Authors fail to thoroughly describe alternative phenotypes of VSMCs in atherosclerotic lesions- not all cells are Macrophage-like.

5. Monocle and NicheNet are analysis tools designed to assist in hypothesis generation- they are not a proof of interaction or differentiation pathways. Experiments would need to be performed to support the conclusions of mechanistic interactions that are proposed in the grant. If experiments are not performed, limitations of the presented approach should be discussed at an appropriate level when describing conclusions.

6. The Trem2 hi macrophage population has no expression (or at least very low) of Trem2. Perhaps this is a miss-leading description of this cluster, or are these a group of genes associated with this cluster that assists naming it?

7. Inconsistent figure quality is a major concern and makes interrogating figures very difficult in numerous instances.

Minor Concerns:

1. Figure II B is very difficult to interpret- the figures appear to be processed differently and maybe have different pixel size.

2. A number of supplemental figures would be useful in the main figures since you can't fully understand the presented figures without the additional data. For example the cluster diversity maps showing the different cluster regions- this is key data to understand how genes are expressed throughout the tissue.

3. Plots in Figure 1 are low resolution and pixelated when zooming in- especially the violin. Please provide higher resolution images to make it easier to examine cluster variation.

4. What is being concluded by the monocle analysis? It may be more informative to examine just smooth muscle derived cells.

5. A series of typos or writing mistakes- Lines 123, 134, 225, 329, etc.

6. What is being compared in Figure 3G?

7. Figure legends are minimal and could be expanded to assist the reader in understanding how data is being presented.

The authors studied VSMC phenotypes and intracellular communication during atherosclerosis progression, by using SCS to compare proximal adjacent and atherosclerotic core regions of human plaques. They identify 8 cell types, amongst which are myeloid cells and VSMCs. Macrophage-like VSMCs seem functionally different between PA and AC according to pseudotime analysis. Transcription factors that contribute to these functional alterations were identified in silico. Moreover, intercellular crosstalk between macrophages and macrophage-like VSMCs was identified.

Data supports most of the conclusions, and the authors address a questions interesting to the field: how/why do these new SMC subsets form. A great image in figure 6A helps the reader decipher the proposed communication .

Nevertheless some major concerns remain, and many details remain to be resolved.

Response: Thanks for the reviewer's positive remarks. We are appreciated to the reviewer's constructive suggestion. We had deeply revised our manuscript according the reviewer's advice. More bioinformatic analyses and experimental data have been added into our result. Some sentences have been revised to make it clarity for the readers. Firstly, the population identities have been validated based on the other datasets through intersection analysis. Secondly, the crosstalk between macrophage and macrophage-like VSMC and the associations of IL-1 β with p-STAT3 and EPAS1 were validated with bone marrow derived macrophages (BMDMs). Thirdly, we also provided the protein levels of main SMC subsets markers/transcription factors, and the inverse correlations of IL-1 β and p-STAT3 in macrophage-like VSMC through *in vivo* experiments.

Reviewer 1:

Major concerns

- **The authors have used a dataset in a public repository, generated by Torkmani et al. and only published as a preprint. Decoding the transcriptome of atherosclerotic plaque at single-cell resolution | bioRxiv . please credit these authors work appropriately.**

Response: Thanks for the reviewer's kindly remind, we had credited the selfless contribution of Mr. Torkamani and his group in the acknowledgement part of our newly-submitted manuscript. And the sentence in the acknowledgement part was written as "And we also give Mr. Torkamani and his group a great amount of credit for their selfless contributing to the single-cell public dataset" at line 587.

- **Annotation: to maintain consistency in the field, it I advisable to compare cell type and subset annotation and markers to previous single cell papers by Wirka (<https://doi.org/10.1038/s41591-019-0512-5>) Fernandez and Pan (<https://doi.org/10.1161/CIRCULATIONAHA.120.048378>). Specifically. Re terminology of SMCs, modified SMCs, fibroblasts, and macrophage subsets.**

Response: Much thanks for the reviewer's advice. To address the reviewer's doubt on re-terminology of SMC and macrophage clusters, we had used several methods to prove the cell type definition in our study. And the detailed reasons were listed as follows. Firstly, because vascular smooth muscle cells (VSMCs) and macrophages involved in atherogenesis were highly heterogenous.^{1,2,3} Two rounds of clustering were applied to fully explore the

heterogeneity of VSMC and macrophage populations in our study. And this method, which define the major cell types in the first-round clustering and specific subtypes in the second-round clustering, is also used by other previous single cell publications in atherosclerosis.^{4,5,6} In Depuydt's and Fernandez's papers, they sub-clustered the major cell types, including VSMCs and immune cells to identify the specific subtypes. Zernecke integrated 9 single cell RNA-Seq studies and then define different leukocyte subsets with sub-clustering method. Secondly, we used the Human Primary Cell Atlas (HPCA) as reference data to annotate our cell clusters through R package SingleR. We found cluster 3, 4, and 15 have higher scores in smooth muscle cell, osteoblast, and fibroblast and cluster 5, 7, 8 have higher scores in monocyte and macrophage (Figure IIA). Furthermore, we conducted the comparisons of top 100 marker genes in different cell clusters in our data with Wirka's, Fernandez's, and Pan's data. However, we can't obtain the cluster marker information of Fernandez's and Pan's dataset as they had not provided in their publications. And we did not receive their reply after sending the help emails to the corresponding authors in their publications (doi: 10.1038/s41591-019-0590-4 and 10.1161/CIRCULATIONAHA.120.048378). Hence, we obtained the top 100 cluster markers of different cell type from Wirka's dataset (Supplementary tables in <https://doi.org/10.1038/s41591-019-0512-5>). And then we conducted the intersection of top 100 gene of our different cluster with Wirka's mouse cell markers. We discovered cluster 3, 4, and 15 share more common markers with SMC and cluster 5, 7, and 8 share more common genes with macrophage (Figure III). Also, cluster 3, 4, and 15 still share more than 25 common genes with SMC subset in Depuydt's data (Figure IIB). Furthermore, we also compared our top 100 markers in different macrophage subsets with Zernecke's single-cell study in order to confirm the validity of sub-clustering (doi: 10.1161/CIRCRESAHA.120.316770). And, we also found the consistency of cell type identities with Zernecke's data (Figure VIB-D). Hence, we supposed that cluster 3, 4, and 15 were belonged to VSMC population and cluster 5, 7, and 8 were belonged to Macrophage population. Thirdly, it's hard to define the specific subclusters in figure 2 as "modulated SMC" in Wirka' publication or "SEM" in Pan's dataset. Because the potential functions and specific markers of these specific cell types have not been fully explored.

• Please rename the major cluster SMCs in 1D, as mesenchymal cells, as this likely also contains fibroblasts and pericytes, as annotated by Wirka. Indeed table 1 shows RGS5 in cluster 3, and DCN/Lum in cluster 4 DEGs. Also, the authors state that in figure 1 clusters express Tagln/Myh11/ACTA2 and annotate them all as VSMCs. These are then reclustered and find fibs? And then in figure 2, they show that, only in cluster 0 the Myh11/Tagln/ACTA2 expression is high, and in the other clusters (1-2-3-4) it is not? But then they proceed to analyze them as "VSMCs"

Response: Thanks for the reviewer's suggestion. As discussed in the last question, we used two round clustering method to identify the specific subsets of major cell type. At the first round of clustering, we observed the highly average expression and percent expressed of smooth muscle cell (SMC) markers (Figure IA-C). The intersection analysis of top makers in cluster 3, 4, and 15 with Wirka's data are more likely to relate to SMC instead of fibroblast or pericyte with statistical significance ($P<0.05$). Secondly, we felt so sorry for leading confusion in figure 2 that the Myh11/Tagln/ACTA2 in cluster 0 expression is high. The

reason why cells in cluster 0 have higher abundance of SMC markers is that we had re-normalized and re-scaled the count data via SCTransform method after using “subset” command for grouping and separation of VSMC and macrophage clusters. This is standardized procedure to conduct sub-clustering.⁷ Accordingly, the highly expression values of SMC markers in cluster 3, 4, and 15 were normalized in our sub-clustering analysis. In Depuydt’s study, we also observed the relatively low expression of SMC markers (ACTA2/MYH11) in S.0 population, which is defined as “Synthetic smooth muscle cells” (Online Figure III).⁵ To avoid misunderstanding this result, we added one more sentence at line 155 as “After separation and grouping with “subset” command, the above procedure was applied in the second round sub-clustering analysis”.

• **W.r.t. sub clustering and pseudotime analysis, please indicate how proliferation and cell cycle were taken into account and show feature plots of major proliferative markers ki67/pcna cell cycle genes or violin plots of signature scores thereof.**

Response: Thanks for the reviewer’s comment. Firstly, there was no significant difference in VSMC subset composition between the PA and AC groups and pathway analysis of dysregulated genes in macrophage-like VSMCs compared with other VSMC subsets indicated contractile and pro-inflammatory functions involvement as we had declared in the result part. Hence, we supposed that VSMCs experience potential functional alterations during atherosclerosis progression. To further validate how functional alterations happened in VSMCs, we conducted the differential expression analysis of VSMCs between PA and AC group (Figure 2G). And the hallmark gene sets of gene set enrichment analysis (GSEA) were related apoptosis, myogenesis, *TGF- β* , and *P53* pathways (Figure IVB). And enrichment analysis of upregulated and downregulated genes in the AC group supported the proliferation and cell cycle in VSMCs through Metascape (<https://metascape.org/gp/index.html>). For example we can observe the enriched pathways in insulin-like growth factor (IGF) transport and uptake and IL-1B signaling pathway in dysregulated genes during disease progression (Figure IVC-D). Another motivation why we explore the differential expression of markers related to cell cycle came from Bennett’s study which reveals *IGF1* can stimulate the VSMC proliferation in the early atherosclerosis and cyclin-dependent kinase inhibitor is involved in VSMC apoptosis with lesions develop.⁸ However, we could not include *TP53*, *MKI67*, *PCNA*, and *CDKN2B* as these genes were not in the top 5000 variable genes. And the violin plots of *IGF1* and *CDKN1A* shown in Figure IVF also confirm the proliferation and cell cycle alterations in VSMCs. We observed the significantly higher expression of *IGF1* (0.01 vs. -0.04) and lower expression of *CDKN1A* (-0.01 vs. 0.05) in PA group. And we also revised the corresponding part in our manuscript. And the sentences were written as “The differential expression analysis of VSMCs between the PA and AC group revealed 323 differentially expressed genes, with 27 genes for the AC group and 296 for the PA group (Figure 2G and Table 6). And the hallmark gene sets of gene set enrichment analysis (GSEA) were related apoptosis, myogenesis, *TGF- β* , and *P53* pathways (Figure IVB), which indicates the proliferation and cell cycle involvement. Furthermore, both enrichment analysis of upregulated and downregulated genes in the AC group supported the functional alterations in VSMCs, including regulation of insulin-like growth factor (IGF) transport and uptake, IL-1B signaling pathway, and etc (Figure IVC and IVD). ~~Furthermore,~~ Hence, we further explored the relative upregulation of *CDKN1A* and downregulation of *IGF1* in the AC group indicated

lowers proliferation and a prolonged cell cycle during atherosclerosis progression (Figure 1 IVE-F)". And we hope these revisions could make our manuscript more clear.

• **Please discuss the evidence for the proposed trajectories in relation to absence of reporter mice. i.e., Wirka analysed enrichment of myh11+ transcriptional profiles in human modified SMCs. This could be done by hypergeometric testing of DEGs in subsets compared to Wirka DEG's.**

Response: Thanks for the reviewer's constructive suggestion. Confronted with technical and ethic restrictions, it's difficult to conduct pseudotime analysis in human VSMCs with the reporter gene expression. According to the reviewer's advice, we extracted the differentially expressed (top 100) genes among different state and finished the intersection analysis with Wirka's mouse cluster markers to validate the relativity of proposed trajectory to SMC (Supplementary figure 1A-E for the reviewer 1). We found that even VSMCs were split into different states by Monocle trajectories, all of states share more common genes with SMC or modulated SMC population with statistical significance (hypergeometric test, P value < 0.0001). Then, we also observed VSMCs in state 1 and 5 are closer to traditional SMC population in Wirka's data as these two clusters are defined at the start part of trajectories (Supplementary figure 1A and 1E for the reviewer 1). And with the pseudotime prolongation, state 2, 3, and 4 are related to modulated SMC (Supplementary figure 1B-D for the reviewer 1). Furthermore, the common genes of state 3 or 4 with modulated SMC are less than that of state 2, which indicated VSMCs in distal path of Monocle trajectories shift away from modulated SMC (Figure VIF). Also, the shift process is also different with the atherosclerosis progression. State 3, which is abundant in AC group, has less common genes than state 4. The results also provide indirectly evidence of the differential state alterations of VSMC in distal path between AC and PA group.

• **How do the authors explain rise of KLF4 TF only in state 4 and 5, not 3, as it was previously identified to act in mf-like SMC differentiation? Shankman et al showed that KLF4 Ko led to less mf-like SMCs (10.1038/nm.3866)**

Response: Much thanks for the reviewer's comment. KLF4 is critically involved in phenotypic transition of VSMCs. In Shankman's research, conditional knockout of KLF4 in SMCs resulted in reduced number of SMC-derived mesenchymal stem cells and macrophage-like VSMCs.⁹ And recent literature integrated the previous VSMC knowledge about KLF4 and shows that KLF4 induces the differentiation of contractile VSMC to mesenchymal, macrophage, fibroblast, and chondrocyte phenotype.¹⁰ Our data also shows that the expression of KLF4 is highly expression in macrophage-like, fibroblast-like, and chondrocyte-like VSMC (Supplementary figure 1G for the reviewer 1). Furthermore, we compared the KLF4 expression between PA and AC group in different VSMC subtypes and found the KLF4 expression in AC group is statistically significantly lower than that in PA group except for macrophage-like subtype (Supplementary figure 1H for the reviewer 1). This result is consistent with lower expression trend of KLF4 in state 3 (Figure 3H). there are several reasons behind this phenomenon. Firstly, KLF4 plays a key role in phenotypic transition within atherogenesis but less research focus on the transcriptional change in VSMCs subtypes in late stage of atherosclerosis.^{9,11} Secondly, this phenomenon is similar as the increased abundance of Tcf21+ cell at 8 weeks and decreased abundance of Tcf21+ cell at 16 weeks in modulated SMC (Figure 1f at Wirka's publication, doi:

10.1038/s41591-019-0512-5). Furthermore, it's interesting that the difference of KLF4 expression in macrophage-like VSMC between AC and PA group has not reached the statistical significance (PA vs. AC: 0.37 vs. 0.35, $P = 0.63$). However, more *in vivo* experimental data is needed in future work. And we had discussed this weakness in our discussion part.

(Supplementary figure 1 for the reviewer 1)

Figure VII: Monocle analysis identified the factional alterations among different states. The intersection analysis of top 100 marker genes in (A) State 1, (B) State 2, (C) State 3, (D) State 4, and (E) State 5 with top 100 marker genes in different cell type identified in Wirka's data. (F) The intersection analysis of top 100 marker genes in different VSMC states with top 100 marker genes in modulated SMC identified in Wirka's data. (G) The violin plot of KLF4 expression among different vascular smooth muscle cells (VSMCs). (H) The violin plot of KLF4 expression in different VSMC subsets split by atherosclerotic (PA and AC) group. Two-sample t test was applied for comparing the KLF4 expression difference; Wilcoxon signed-rank test for comparing intersection genes difference. * $P < 0.05$, ** $P < 0.01$, *** $P < 0.001$, **** $P < 0.0001$.

• All human data is based on RNA level, partly derived from digested tissue, and hence, please provide validation on protein level and/or in intact carotid artery segments using

immunohistochemistry and/or western blotting for main SMC subsets markers/transcription factors, and the inverse correlations of Il1b and STAT3.

Response: Much thanks for the reviewer's advice. However, it was very difficult to conduct the protein level validation in the human atherosclerotic tissue in our hospital as the Ethics Committee is strict in specimen acquirement. Accordingly, the artery segments from apolipoprotein E deficient (*ApoE*^{-/-}) mice fed with a high-fat diet (HFD) were used to provide validation on protein level of main SMC subsets markers/transcription factors, and the inverse correlations of IL-1 β and STAT3. Firstly, the aorta of *ApoE*^{-/-} mice with HFD were collected to conduct western blot assays (Supplementary figure 2 for the reviewer 1). Quantification analysis indicated the upregulated expressions of CD68 (macrophage-like VSMC marker), FN1 (Fibroblast-like VSMC marker), and SOX9 (Chondrocyte-like VSMC marker) and downregulated expressions of p-STAT3, and α -SMA (VSMC marker) during disease progression (16 weeks vs. 8 weeks) (Figure 8A-D and X). And we also observed the downregulation of p-STAT3 can be reversed by the adeno-associated virus-IL-1 β (AAV-sgIL-1 β) (Figure 8D). But the expression of CD 34, the marker of endothelial cells, did not change with disease progression and AAV-sgIL-1 β intervention (Figure X). Secondly, immunofluorescence staining experiments have been finished to validate the inverse association of IL-1 β and STAT3.

AS 8w	+	-	-
AS 16w	-	+	+
AAV-sgIL-1 β	-	-	+

(Supplementary figure 2 for the reviewer 1)

- **Please repeat main experiments with primary mouse macrophages such as bone-marrow derived or naïve peritoneal macrophages, or human PBMCs. Please note, thioglycolate-induced peritoneal macrophages are not naïve/resident like.**

Response: According to the reviewer's advice, we had repeated the main experiments with the bone marrow derived macrophages (BMDMs) instead of the RAW264.7 cells. After the ox-LDL treatment, the level of IL-1 β is also increased in macrophage medium (MM) (Figure IX). Furthermore, we also observed the macrophage-like trans-differentiation process, downregulation of p-STAT3 and upregulation of EPAS1 at 48 h with the MM treatment (Figure IXB-E). And also, downregulation of p-STAT3 and upregulation of EPAS1 regulated by IL-1 β can be reversed by IL-1Ra (Figure IX).

- **The statement in line 386 that EPAS1 and P-STAT3 are directly related to Il1b cannot be justified by the data of figure 6. Please adjust statement. For P-stats3 this is shown in figure 7. oxLDL initiates many affects, a.o. reactive oxygen species, also able to stabilize EPAS1 levels. Please silence or block IL1b in the presence of oxLDL to support the direct relationship.**

Response: Much thanks for the reviewer's kind advice. We also found the inappropriate statement that EPAS1 in macrophage-like VSMCs is regulated by IL-1 β . Hence, we had conducted the western blot assays of EPAS1 with the loss function of IL-1 β after MM treatment (Figure IXE). And we observed that the upregulation of EPAS1 is reversed by IL-Ra.

• Please avoid the term “ageing” to refer to the transition between early and late stages in the differentiation process of Mf-like SMCS.

Response: Much thanks for the reviewer's suggestion. Aging include many cell damages and phenotypic changes. So, it could be inappropriate to use the term “aging” to refer to transition between early and late stages in the differentiation process of Macrophage-like SMC. Accordingly, we used the “functional alteration” to replace the term of “aging” in our manuscript.

Minor

Abstract:

- Line 35: intracellular (signalling, within cell) or intercellular (crosstalk)?

Response: We had changed “intracellular” as “intercellular” as the reviewer's advice.

- line 37 please include source and artery

Response: According to reviewer's advice. We had re-written this sentence as “Single-cell RNA-sequencing (scRNA-seq) of proximal adjacent (PA) and atherosclerotic core (AC) regions of human carotid artery plaques was employed to identify the distinct cell compositions during atherosclerosis progression”.

- In intro/abstract, the authors focus on VSMCs and their associated cellular crosstalk very much. However eventually, they focus more on the effect of other cells (macrophages) on the VSMCs.

Response: Much thanks to reviewer's suggestion. Our focus is the VSMC's functional alterations and potential cellular crosstalk with macrophage in this manuscript. And we found macrophage is deeply involved in VSMC's functional alterations as our results showing. Hence, we eventually discussed more about the effect of other cells (macrophages) on the VSMCs and VSMC's intracellular pathways in abstract and discussion part. To avoid misleading in our abstract, we had re-written the first sentence in abstract as “Vascular smooth muscle cells (VSMCs) play a central role in atherosclerosis progression, but the functional changes in VSMCs and the associated cellular crosstalk, especially the interactions with macrophages, that contribute to atherosclerosis progression remain unknown”. We hope this change can help the manuscript clarity.

Introduction

- Writing not always great (line 111: doesn't flow well, line 112: starting sentence with “But”, Line 114 and 117: comma missing, line 122)

Response: We had re-written the sentence in line 111 as “For example, macrophage-like VSMCs are likely contributed to plaque instability, in light of their pro-inflammatory profile”. And the sentence in line 112 had been changed as “However, current research indicates that macrophage-like VSMCs are functionally different from myeloid-derived macrophages”. Also, we had added the comma in line 114, 117, and 122.

- A little unclear whether focus on macrophage-like VSMCs, or the crosstalk between macrophages and VSMCs, or both? Hypothesis? Will the macrophage-like VSMC interact with macrophages, or what do you expect?

Response: Much thanks for the reviewer's advice. We focused on the functional alterations of VSMCs and the crosstalk between macrophage-like VSMC and macrophage during atherosclerosis progression in our manuscript. And we had revised the last two sentences in abstract as "VSMC functional alterations and crosstalk between different subtypes of VSMCs and macrophages were also been explored. Then regulatory effect of macrophages on macrophage-like VSMCs was further examined through *in vitro* and *in vivo* tests".

- line115-117: please include "have been identified in human atherosclerotic plaques". And include types described in mice (Zernecke et al, meta-analysis.)

Response: Thanks for the reviewer's advice. We had added "human" in the line 115-117. Also, we had added another sentence as "And, two more macrophage subset, including IFN γ and cavity macrophages have been revealed by a meta-analysis based on mouse data". Furthermore, we had revised the last sentence of this paragraph as "The inflammatory macrophages identified from both human and mouse data, forming the major macrophage population within the plaque intima, are enriched for numerous pro-inflammatory factors" to make this paragraph flow well.

Methods

- Several textual mistakes/unclarities: line 134, line 143, line 148, 174, 225, 227, 242

Response: We had corrected the textual mistakes in line 134, 227, and 242. And, the sentence in line 143 had been revised as "Finally, a total of 3, 5021 individual cells were included in subsequent analysis". We also added another sentence in line 148 as "Clustering tree is useful for visualizing the relationship between clusters at multiple resolutions". The sentence in line 174 had been re-written as "We then used NichNet analysis to predict the interactions between ligands secreted by macrophage and targets in VSMC subsets." And the sentence in 225 had been written as "Cells and tissue were lysed in lysis buffer for 60 min, followed by centrifugation at 12,000 \times g for 20 min".

- Very detailed (scRNA-seq analysis), methodology seems correct and applicable

Response: Much thanks for the reviewer's remark. And we had added another sentence in line 150 as "Then, intersection analysis between top markers of different clusters in our study with published data were used to validate the cluster identities" according to the reviewer's advice on intersection of wirka's data.

- Line 209: C57BL/6N or C57BL/6J?

Response: We used C57BL/6J mice in our study. And we had corrected this mistake for the whole manuscript.

- Explain why only male mice were used for in vitro experiments. Why not use human cells? For further validation of in silico data in human situation.

Response: The reason why we only selected the male mice in our experiments is that the effects of estrogen on artery disease are protective and positive.^{12, 13} Hence, female mice were excluded from the study to avoid potential hormones' effect. Secondly, the absence of cells from human *in vitro* experiments is indeed a weakness of this study. The primary human artery smooth muscle cells (HASMCs) could be obtained from company or extracted from patients undergoing surgery in hospitals. However, it's almost impossible to extract HASMCs

from patients in our hospital because of ethic restrictions. And, purchasing primary HASMCs from company has been held up according to the COVI-19 pandemic. Furthermore, the effect of disease conditions in HASMCs from different cell lines could influence the experimental results in our study as the reviewer has mentioned. Hence, the primary VSMCs from C57BL/6J mice artery is the better choice to validate the silico analysis. Also, we also finished the *in vivo* experiments in order to make our conclusion more convincing. And we hope the new revisions could reach your requirements.

- Gender of human patients not stated

Response: Much thanks for reviewer's suggestion. The data in GEO database did not provide the gender information of these samples. And we also read the authors' publication at bioRxiv website. And we did not find out the supplementary table 1, which provides the patient's characteristics. No reply has been received after contacting with the authors. Hence, it's a pity that we could not state the gender of the human patients.

- Line 136-8: what is meant by "The microarray gene expression profile 137 (GSE43292) and corresponding Stary classification of plaques were also downloaded from the 138 GEO dataset" is this the same patients or others.

Response: This sentence indicates that the microarray gene expression profiles and corresponding plaque severity classification information have been download from the GEO database. To make it clarity, we had revised this sentence as "The microarray gene expression profile (GSE43292) and corresponding plaque severity classification information were also downloaded from the GEO database for subsequent bulk RNA analysis".

- Line 236; how was oxLDL generated or what was its source?

Response: Much thanks for the reviewer's advice. Ox-LDL are come from human source in our study. Ox-LDL were purchase from Yiyuanbiotech, the largest research institute for separation and purification of plasma lipoprotein in China. Secondly, ox-LDL separation steps are shown as follows:

Isolation: LDL is isolated from blood bank produced human plasma, via ultra-centrifugation (1.019-1.063g/cc). It is purified via ultracentrifugation homogeneity determined by agarose gel electrophoresis. The LDL is membrane filtered and aseptically packaged, store at 4°C.

Oxidization: Human LDL is oxidized using Cu2SO4 (oxidant) in PBS. Oxidation is terminated by adding excess EDTA-Na2. Each lot is analyzed on agarose gel electrophoresis for migration versus LDL. This lot of oxLDL migrates 2.0-fold further than the native LDL.

(Supplementary figure 3 for the reviewer 1)

Yiyuanbiotech Native-LDL(n-LDL), Acetylated-LDL(Ac-LDL) and Oxidized-LDL(ox-LDL) were loaded on a agarose gel and electrophoresed for 60 minutes. The lipoproteins were stained with Sudan Black (A and B). Oil red O staining was used to determine the formation of foam cell. RAW264.7 were incubated with 50µg/mL ox-LDL for 24hrs (C) (Source: Used with permission of Yiyuanbiotech)

Thirdly, there is no murine ox-LDL on the market at present in China. Considering that ox-LDL has a short validity period and COVID-19 pandemic (4°C, 1 month), it's not appropriate for us to purchase from abroad. Studies have reported that human-derived ox-LDL can also establish a stable and reliable model of atherosclerosis in mice *in vitro*, due to the high species homology of LDL.^{14, 15, 16}

Results

- Results difficult to follow at times

Response: Thanks for the reviewer's advice. We had revised our results strictly according to the reviewer's questions asked below. Firstly, we had added several figures to support our conclusions. Secondly, we also revised our sentences in the result part to make the logic clearer. And we hope the newly-submitted manuscript could reach to the reviewer's requirements.

- Figure 1E: Add individual datapoints, and number of cells/group.

Response: We had added the individual datapoints in Figure 1E as reviewer's advice. Although we had made the point size as small as possible, it would be still a little difficult to observe the points as the large number of cells per cluster. And we also had added the information of cell numbers per cluster as table 2.

- Figure 2F does this represent DEGs across all 4 subsets? It seems more insightful to also show DEGs specific for each subset?

Response: Figure 2F represent the top pathways in macrophage-like subsets across different groups (PA vs. AC). Because we were much more focused on macrophage-like in our manuscript. And Figure 2G shows the dysregulated genes between PA and AC group. We also conducted the functional enrichment analysis across all 4 subsets have been shown in Figure IVB-D.

- Several textual mistakes/unclarities: line 252, 313, 362, 363

Response: The "*PDGERB*" in line 252 had been changed as "*PDGFRB*". The sentence in line 313 has been expanded into two sentences as "Accordingly, we conducted BEAM analysis in order to identify genes associated with the second branch node. And then, the genes with branch-dependent expression were assigned them to 3 gene clusters according to trajectory differentiation to identify the mechanism by which the VSMC fate decision is made". And the "factions" in 362 had been revised as "fractions". And to make the sentence clarity in line 363, we had deleted the word "different" and "type". And the sentence had been revised as "we observed the main fractions of immune cell were macrophage subsets (macrophage M0 and M2), which were significantly activated in atheroma plaque group".

- Line 251: state more clearly that info can be found in supplements (Table 1, but applies to whole manuscript)

Response: Much thanks for the reviewer's advice. We had revised the simple sentence in line 251 as "The differently expressed marker genes of each cluster listed in Table 1 were used for cluster annotation".

- Line 288-292: you speak of cluster 0, 2 and 3, whereas in the figure, you speak of cluster 0, 1 and 2 ◊ confusing, please change

Response: Thanks for the reviewer's advice. We had corrected this mistake. The "cluster 2" has been revised as "cluster 1", and "cluster 3" has been revised as "cluster 2".

- Figure 4A, please also provide this list as a table to allow colleagues to re-use data, as font in 4A is very small.

Response: Thanks to the reviewer's suggestion. It's hard to enlarge the font size in Figure 4A. And we had provided a table of Figure 4A, including the transcription factor information and corresponding viper scores in table 11.

- In general: switch PA and AC results around (PA first in graphs, then AC)

Response: According to the reviewer's advice. We had switched PA at first place in our manuscript. "AC vs. PA" or "atherosclerotic core (AC) and proximal adjacent (PA)" had been revised as "PA vs. AC" or "proximal adjacent (PA) and atherosclerotic core (AC)".

- Line 331-332: The consistency of TF activity trends in different atherosclerotic groups (PA vs. AC) and states further confirmed the central role of macrophage-like VSMCs during atherosclerosis progression. ◊ based on in silico analysis only. Further validation is required.

Response: Much thanks for the reviewer's advice. It's inappropriate to use "confirm" in line 331-332 as no further experiments to validate. Hence, this sentence had been revised as "The consistency of TF activity trends in different atherosclerotic groups (PA vs. AC) and states further indicted the central role of macrophage-like VSMCs during atherosclerosis progression".

- Figure 5: in text: the top receptors in VSMCs including ACKR4, IL1R1 and JAM3 are involved in inflammatory processes. Hence, we focused on the interaction of macrophage-like VSMCs with other subsets of macrophages and VSMCs in the PA and AC groups. ◊ Did you also analyze ligand-receptors of the macrophage-like VSMCs specifically?

Response: Yes, we also analyzed the ligand-receptors of macrophage-like VSMCs specifically. And the result was shown in Figure VIII A. And we also found that IL1R1, JAM3 are also top receptors in macrophage-like VSMC.

- Authors assessed ligands that contribute to the transcriptional response in macrophage-like VSMCs. Vice versa, do these macrophage-like VSMCs also affect the transcriptional response of other plaque cells?

Response: Much thanks for the reviewer's advice. It's also interesting to explore the influence of macrophage-like VSMC on the other plaque cells. This is also our group plan in future work, especially the effect of macrophage-like VSMC on macrophage. Another protein IRF1 may played a key role in this process. However, we need more data and experiments to support this potential relationship. And we wish to discuss this relationship in future study.

- Authors should more clearly explain why STAT3 and IL1b are chosen to validate further. Is STAT3 a target of IL1b?

Response: Thanks for the reviewer's advice. There are several reasons why we chose IL-1 β and STAT3 in our study. Firstly, we observed the highly average expression and percent expressed of IL1B as the ligand of myeloid cells on VSMCs. And in the further subset cell crosstalk analysis, we observed IL1B is still highly expressed in inflammatory macrophages

and further upregulated in AC group (Figure 5A, 5C-D). Secondly, previous research has highlighted the clinical significance of IL-1 β during atherosclerosis progression as we had discussed in the discussion part. Thirdly, bioinformatics results suggest that STAT3 is a potential target of IL-1 β . Literature reveals that STAT3 plays an important role in the progression of AS, but its specific mechanism has not been proven and its function is controversial,^{17, 18, 19} so we chose to explore IL-1 β and STAT3. And we also added one sentence at line 412 as “According to the dominant role of IL-1 β involved in macrophage-macrophage-like VSMC crosstalk, we chose IL-1 β and STAT3, one of its targets with pleiotropic effect in atherosclerosis, for the further experimental validation”.

- Figure 5D font size is not readable

Response: we had enlarged the font size to make it readable according to reviewer’s advice.

- Line 360-2 figure 8 should be figure 6,

Response: Much thanks for the reviewer’s kind suggestion. We had revised “Figure 8” in line 360-2 as “Figure 6”.

- Please introduce in the results how the immune cell fractions were (acquired i.e., CIBERSort)

Response: We had added one more sentence in front of immune cell fraction in the result part as “Furthermore, the immune cell type proportions were estimated by CIBERSORT based on bulk tissue gene expression profiles”. And it would be more helpful for the readers to know the immune cell fractions.

- Line 363: why subdivision into M0/M1/M2 macrophages here, whereas TREM2hi/pro-inflammatory/resident earlier?

Response: Much thanks for the reviewer’s question. The CIBERSORT algorithm only provides M0/M1/M2 macrophage in its analysis. But it’s reasonable to divide the macrophages into Trem2-hi/pro-inflammatory/resident subsets in atherosclerotic plaque. However, few publications have explained the correlations between M0/M1/M2 and Trem2-hi/pro-inflammatory/resident at now. In Willemsen’s review, resident-like and Trem2-hi macrophages may resemble an M2-like phenotype and inflammatory macrophage may be associated with an M1-like phenotype.²⁰ But further study is needed for validation. Our aim is to provide a new insight of the relationships among IL-1 β , STAT3 and immune cells.

- Figure 6E placement is confusing, possibly flip orientation and place below B-D. in the legends explain color-coding of red and blue, and what it means if no coefficient is shown.

Response: Much thanks the reviewer’s advice. We had changed the layout of Figure 6 to make it more clarity. Also, we had added instructions of red/blue in figure 6E, which has been shown as “red means positive correlation, while blue means negative correlation” in figure legends.

- 6 The figure is unclear \diamond IL1b is inversely related to the proportions of mf subsets, and STAT3 positively correlated with proportions of mf subsets. \diamond isn’t IL1B positively correlated, and STAT3 inversely?

Response: Much thanks for the reviewer’s question. Inflammatory macrophage may be associated with an M1-like phenotype as we discussed above. And we observed that *IL-1 β* is positively correlated with M1 macrophage proportion and *STAT3* is negatively associated

with M1 macrophage proportion in figure 6E, which is consistent with the reviewer's opinion. Furthermore, *IL-1 β* is also positively associated with M0 and M2 macrophage. And STAT 3 is negatively associated with M2 macrophage.

- Line 367-370: "Accordingly, we propose that IL-1B secreted by inflammatory macrophages induces the functional alterations in macrophage-like VSMCs through downregulating STAT3, and this signaling cascade is the main driver of atherosclerosis progression." \diamond this is only based on in silico analysis and correlations. So please adjust the statement to avoid overinterpretations

Response: Much thanks for the reviewer's advice. We had revised this sentence as "Accordingly, we propose that IL-1 β secreted by inflammatory macrophage induces the functional alterations in macrophage-like VSMCs through downregulating *STAT3*, and this signaling cascade may play as a main driver in atherosclerosis progression".

- Figure 7: why use mouse macrophages? Why macrophage-like VSMC "aging"?

Response:

Response: Much thanks for the reviewer's advice. The reason why we use mouse cells has been explained as above. The mouse derived primary macrophages and VSMCs were the optimal choice in our *in vitro* experiments. And we had realized the inappropriate use of "aging" in our manuscript. Hence, "aging" was replaced as "functional alterations" in the whole manuscript.

- Line 378; explain "RM" here too

Response: According to reviewer' advice. We had explained RW in line 378 as "RAW264.7 medium".

- Line 389: what is the function of IL-Ra, static, and Colivelin? This is not mentioned in methods. In the methods its name is IL-1Ra

Response: Much thanks for the reviewer advice. We had added the descriptions of Static (the inhibitor of STAT3) and Colivelin (the activator of STAT3) in the method part. Furthermore, we had revised "IL-Ra" as "IL-1Ra" in the result part.

- Line 391: how did the authors acquire early and late STAGE Mf-like VSMCs?

Response: Much thanks for the reviewer's question. The VSMCs treated with RM at 12h and 48h are defined as early and late stage of macrophage-like VSMC, respectively. And, we had revised this sentence as "Furthermore, we compared functional alterations between early- and late- stage macrophage-like VSMCs, which has been treated with RM at 12h and 48h, respectively" to make it clarity. Also, we had added one sentence in the method part as "Early and late stage macrophage-like VSMCs were defined as co-culturing with RM/MM for 12 hours and 48 hours, respectively" for a better understanding.

- Figure 7 please indicate number of biological (mice) and technical replicates (wells per assay condition), as well as number times the expt was repeated in the legend.

Response: Thanks for the reviewer's advice. Technical repetition is simply the repetition on a single individual to ensure an accurate measurement of the individual sample. In this study, the technical repetition was 3 to ensure the stability of individual sample. Biology samples from different biological sources, that is, each measurement is based on a sample from a different individual, making the results universal. The number of mice in each group in this study was 6, and the wells for cell experiments was 4-8 per group. And the number of replicates in each group has been shown in the figure legends.

Enlarge size of scatter in plot: signs are too small to see replicates. Please adjust layout of figure 7 to avoid confusion: swap quantification of 7A with 7C; o second row show 7B, 7C western and 7c quantifications. Indicate statistics for multiple groups shown in figure 7, and confirm normal distribution of data to permit T-test instead of non-parametric

Response: Much thanks the reviewer's advice. We had enlarged the size of points in figure 7. Also, we had changed the order of different subplots according to the reviewer's suggestion. And we wish it would be clearer to read. Furthermore, we also examined difference between groups with Mann-Whitney tests. And the differences between groups still remain the statistical significance.

Reference

1. Grootaert MOJ, Bennett MR. Vascular smooth muscle cells in atherosclerosis: time for a re-assessment. *Cardiovascular research* **117**, 2326-2339 (2021).
2. Cochain C, *et al.* Single-Cell RNA-Seq Reveals the Transcriptional Landscape and Heterogeneity of Aortic Macrophages in Murine Atherosclerosis. *Circulation research* **122**, 1661-1674 (2018).
3. Lin JD, *et al.* Single-cell analysis of fate-mapped macrophages reveals heterogeneity, including stem-like properties, during atherosclerosis progression and regression. *JCI insight* **4**, (2019).
4. Zernecke A, *et al.* Meta-Analysis of Leukocyte Diversity in Atherosclerotic Mouse Aortas. *Circulation research* **127**, 402-426 (2020).
5. Depuydt MAC, *et al.* Microanatomy of the Human Atherosclerotic Plaque by Single-Cell Transcriptomics. *Circulation research* **127**, 1437-1455 (2020).
6. Fernandez DM, *et al.* Single-cell immune landscape of human atherosclerotic plaques. *Nature medicine* **25**, 1576-1588 (2019).
7. Rao1 M, *et al.* Resolving the intertwining of inflammation and fibrosis in human heart failure at single-cell level. *Basic Research in Cardiology*, (2021).
8. Bennett MR, Sinha S, Owens GK. Vascular Smooth Muscle Cells in Atherosclerosis. *Circulation research* **118**, 692-702 (2016).
9. Shankman LS, *et al.* KLF4-dependent phenotypic modulation of smooth muscle cells has a key role in atherosclerotic plaque pathogenesis. *Nature medicine* **21**, 628-637 (2015).

10. Yap C, Mieremet A, de Vries CJM, Micha D, de Waard V. Six Shades of Vascular Smooth Muscle Cells Illuminated by KLF4 (Kruppel-Like Factor 4). *Arteriosclerosis, thrombosis, and vascular biology* **41**, 2693-2707 (2021).
11. Alencar GF, *et al.* Stem Cell Pluripotency Genes Klf4 and Oct4 Regulate Complex SMC Phenotypic Changes Critical in Late-Stage Atherosclerotic Lesion Pathogenesis. *Circulation* **142**, 2045-2059 (2020).
12. Miller VM, Duckles SP. Vascular actions of estrogens: functional implications. *Pharmacological reviews* **60**, 210-241 (2008).
13. Vitale C, Mendelsohn ME, Rosano GM. Gender differences in the cardiovascular effect of sex hormones. *Nature reviews Cardiology* **6**, 532-542 (2009).
14. Zhang Y, *et al.* Irisin Inhibits Atherosclerosis by Promoting Endothelial Proliferation Through microRNA126-5p. *Journal of the American Heart Association* **5**, (2016).
15. Yang J, *et al.* Store-operated calcium entry-activated autophagy protects EPC proliferation via the CAMKK2-MTOR pathway in ox-LDL exposure. *Autophagy* **13**, 82-98 (2017).
16. Du RH, *et al.* Fumigaclavine C activates PPAR γ pathway and attenuates atherogenesis in ApoE-deficient mice. *Atherosclerosis* **234**, 120-128 (2014).
17. Chen Q, *et al.* Targeted inhibition of STAT3 as a potential treatment strategy for atherosclerosis. *Theranostics* **9**, 6424-6442 (2019).
18. Dutzmann J, Daniel JM, Bauersachs J, Hilfiker-Kleiner D, Sedding DG. Emerging translational approaches to target STAT3 signalling and its impact on vascular disease. *Cardiovascular research* **106**, 365-374 (2015).
19. Yang L, *et al.* PM(2.5) promoted lipid accumulation in macrophage via inhibiting JAK2/STAT3 signaling pathways and aggravating the inflammatory reaction. *Ecotoxicology and environmental safety* **226**, 112872 (2021).
20. Willemsen L, de Winther MP. Macrophage subsets in atherosclerosis as defined by single-cell technologies. *The Journal of pathology* **250**, 705-714 (2020).

Reviewer #2 (Remarks to the Author):

In the paper by Xue et. al., the authors use single cell gene expression analysis of human atherosclerotic plaque or adjacent proximal regions to better predict potential regulatory interactions between key cell types in disease progression. They use this data to describe clusters associated with different regions of the aorta to focus primarily on smooth muscle cell diversity. Furthermore, they use prediction approaches to propose potential interactions between essential cell types known to function in atheroma formation, specifically macrophage and smooth muscle cell. Their final conclusions are that macrophages regulate smooth muscle function in atherosclerosis, but data supporting this conclusion is limited beyond predictive algorithms.

The study design is innovative and of potential interest to the field. However, a number of major concerns exist regarding data analysis, presentation, and interpretation. The primary concern is the over interpretation of results, in particular drawing conclusions from informatics approaches without showing sufficient follow-up experiments. And second, potential contradictions in the data presentation that make it difficult to determine what is being presented and whether the conclusions being presented are relevant to disease. A number of specific requests are outlined below that may improve the clarity of the paper and make it more accessible to potential readers.

Response: Much thanks for the reviewer's remarks. We felt grateful to revise our manuscript according to the proposed questions, as these suggestions are meaningful and helpful to our manuscript. Firstly, we had adjusted the overinterpretation statements in our manuscript. And, more supportive data pertaining the *in vivo* and *in vitro* experiments have been added in the newly-submitted manuscript. We hope our statements could be reasonable with these data. Secondly, the figures and corresponding descriptions had been revised for more accessible to readers. The figure 1-2, 5-7 have been re-plotted, we had enlarged the font size, changed the order of sub-plots, and added the datapoints in violin plot in these figures. Furthermore, we added figure 8, II-III, figure IVD, figure IVF, figure VIB-D, Figure IX-X to support and validate our conclusions. Also, the corresponding descriptions in the result part and figure legends have also been revised. We hope these revisions could meet the reviewer's requirement.

Major Concerns:

1. CD68 expression appears to be absent in smooth muscle cell clusters in Figure I and Supplemental Fig I, but in subsequent figures CD68 expression is a key defining marker for a smooth muscle cell population of interest. This needs further explanation to understand where the smooth muscle cell cluster is deriving from.

Response: Much thanks for the reviewer's question. We had conducted two-round of clustering procedure in our analysis. The purpose of first round of clustering is to identify the major cell type in our single cell data. So, cluster 5, 7, and 8 with highly expression of CD68 and CD14 are defined as myeloid cells and cluster 3, 4, and 15 with highly expression of TAGLN and ACTA2 are recognized as vascular smooth muscle cell (VSMC) at the first round of clustering. To further validate the rationality of the cell type definition, we have compared the top 100 differently expressed genes of the different clusters in our data with Wirka's top 100 markers of different cell type. We discovered cluster 3, 4, and 15 share more

common markers with SMC and cluster 5, 7, and 8 share more common genes with macrophage (Figure III).¹ The result also validates the identities of cluster 3, 4, 15, cluster 5, 7, and 8. Then the count data has been re-normalized and re-scaled via SCTransform method after using “subset” command for separation of VSMC and macrophage clusters at the second round of clustering, respectively.² Hence, the highly expression markers of VSMC and macrophage have been rescaled to further explore the heterogeneity in the major cell type. This method is useful to identify the different phenotypes of the specific cell type and has also been implemented in atherosclerosis.^{3, 4, 5} According to the relatively high expression of CD68 and CD74 in cluster 1 and 4, we defined these cells as macrophage-like VSMC.⁶ To avoid misunderstanding this result, we also we added one more sentence at line 155 as “After separation and grouping with “subset” command, the above procedure was applied in the second round sub-clustering analysis”. Furthermore, as Zernecke’s meta-analysis had concluded the top 100 marker genes of different macrophage subtypes, the intersection analysis of top 100 marker genes in the different macrophage clusters with those in Zernecke’s data had been applied to validate the sub-clustering validity based on two-round of clustering procedure (Figure B-D).

2. It was surprising to me that clusters (particularly macrophage) representation didn’t appear to be dramatically different between the sources of cells isolated. Could the authors explain this observation, since it is contradictory to what one might typically expect to fine.

Response: Much thanks to the reviewer’s advice. Several reasons for the similarity of different phenotypes of macrophage between PA and AC group are listed as follows. Firstly, the single cell data are high-dimensional, noisy, and sparse due to the current scRNA-seq technologic characteristics. And the single cell data has been dimensional reduced via principle component analysis and then grouped based on patterns embedded in gene expression by Seurat package.⁷ Accordingly, the cells clustered together are considered as the same cell type. Secondly, PA and AC group are recognized as the early and late stage of atherosclerosis instead of normal and diseased group. And, macrophage has been trans-differentiated into three canonical macrophage subsets once atherogenesis process initiation.⁸ Furthermore, three canonical macrophage subsets in early and late stage of atherosclerosis or progression and regression group of atherosclerosis have also been observed in Zernecke’s publication, which also validate our viewpoint (Figure 3 in Zernecke’s data).⁵ Thirdly, the abundances of three macrophage subsets between PA and AC group are various in our data. The TREM2^{high} subset recognized as foamy cell composes 40% of macrophages in AC group but 19% of macrophages in PA group indicative of the cholesterol deposition in macrophage with atherosclerotic progression. And we hope these explanations could help the reviewer to understand this result in our data, especially figure V.

3. The authors use “macrophage-like” in a number of contexts to describe a smooth muscle cell cluster type. While this nomenclature has been used sporadically in previous publications, there is little data provided beyond CD68 gene expression to suggest similarity between these cells and true macrophages. Perhaps the cells may be more accurately described as CD68+ smooth muscle cells- since this is all that is shown in the manuscript. If the authors performed more in depth comparative analysis or even functional experiments, it may become reasonable to call the cluster “macrophage-like”.

Response: Much thanks for the reviewer's advice. It is not equal to describe the CD68+ VSMC as macrophage-like VSMC. However, there exists a gap between the functions and nomenclature of the macrophage-like VSMC in VSMC subtype research. Current reviews conclude several characteristics of macrophage-like VSMC in atherosclerosis based on the VSMCs with high expression of markers and key transcriptional factors (including CD 68, LGALS3, MCP1, and etc).^{6, 9, 10, 11} However, other research indicated that macrophage-like VSMCs are functionally different from myeloid-derived macrophage in culture.^{12, 13} Hence, the specific functional experiments for macrophage-like VSMC in atherogenesis and atherosclerotic progression have not been explored. Furthermore, we also observed the activation of inflammation, adhesion, and apoptosis in CD68+ VSMC from Figure 7, which also validates the CD68+ VSMC's trans-differentiation tendency to macrophage-like state. Moreover, the identification of the VSMC with CD68+ as macrophage-like VSMC have also been applied by previous publications.^{14, 15} Hence, we still adapted this nomenclature in our manuscript.

4. Authors fail to thoroughly describe alternative phenotypes of VSMCs in atherosclerotic lesions- not all cells are Macrophage-like.

Response: Much thanks for the reviewer's comment. The other phenotypes of VSMC also play an important role in atherosclerosis. For example, osteogenic VSMC likely to contribute to plaque calcification and plaque destabilization.¹⁶ However, the main conclusion in our research focused on macrophage-like VSMCs with crucial role during atherosclerosis progression and enhanced apoptosis, inflammation, and adhesion at late stage induced by inflammatory macrophage based on our bioinformatic analysis and experimental data. The reasons why we chose macrophage-like VSMC to further explore in our study were listed as follows. Firstly, we conducted the differentially expressed analysis of VSMC between PA and AC group. The top differently expressed genes are associated with lipid metabolism (*APOD* and *APOE*), inflammation (*CXCL14*, *CCL4* and *SPPI*) (Figure 2G). Furthermore, the hallmark gene sets of gene set enrichment analysis (GSEA) were related apoptosis, myogenesis, *TGF-β*, and *P53* pathways (Figure IVB). And the pathway analysis (KEGG) of dysregulated genes in macrophage-like VSMCs is also related to contractile and pro-inflammatory functions (Figure 2F). Accordingly, we supposed that macrophage-like phenotype involved in functional alterations of VSMC during disease progression. Secondly, the distal path, which mainly contained macrophage-like VSMCs (Figure 3C-D), was mainly state 4 in the PA group and state 3 in the AC group, which indicated the state alterations of VSMCs during atherosclerosis progression. Hence, we are more interested in macrophage-like VSMC, which has also been reported involvement in inflammatory process in other studies.^{14 17} The following cell-to-cell communication analysis and *in vitro* tests further validate the crosstalk of macrophage with macrophage-like VSMCs through IL-1β-STAT3 axis. Furthermore, we had conducted experiments in mouse models to partly address the reviewer's question. *In vivo* analysis in our study also indicated that VSMCs would transdifferentiate into fibroblast-like (FN1), macrophage-like (CD68), and chondrocyte-like phenotypes (SOX9) with the atherosclerosis progression (16 vs. 8 week) in Figure 8 and X.

5. Monocle and NicheNet are analysis tools designed to assist in hypothesis generation- they are not a proof of interaction or differentiation pathways. Experiments would need

to be performed to support the conclusions of mechanistic interactions that are proposed in the grant. If experiments are not performed, limitations of the presented approach should be discussed at an appropriate level when describing conclusions.

Response: Thanks for the reviewer's advice. It's may not enough convincing to propose the cell interactions in our data without adequate experimental data (Figure 6A). We only identified the crosstalk of macrophage-like VSMC with macrophage through IL-1 β -STAT3 axis through following *in vivo* and *in vitro* experiments. The other cell interactions with macrophage-like VSMC have not been validated through experiments. Accordingly, we had validated the crosstalk between macrophage and macrophage-like VSMC with primary macrophage. Also, we had added one more paragraph to declare the limitations of our study to address the reviewer's concerns. And the sentences have been written as "However, few limitations in our study should be declared. Firstly, although we had proved the regulatory relationship of TFs (including EPAS1 and STAT3) with IL-1 β during inflammatory macrophage-to-macrophage-like VSMC interactions through *in vitro* analyses, the other relationship between ligands and target TFs, cell interactions with macrophage-like VSMC should be further validated in the future work".

6. The Trem2 hi macrophage population has no expression (or at least very low) of Trem2. Perhaps this is a miss-leading description of this cluster, or are these a group of genes associated with this cluster that assists naming it?

Response: Thanks for the reviewer's comment. We felt so sorry for lack of enough description in TREM2-high cluster definition. Indeed, we defined the cluster 0 as TREM2-high phenotype based on a group of genes. Except *TREM2*, other markers of TREM2-high subtype *CD9*, *LGALS3*, *FLOR2*, and *ABCA1* are highly expressed in cluster 0. ^{8, 18} Furthermore, we also conducted the intersection analysis of top markers in our different macrophage clusters with Zernecke's leukocyte subsets (Figure VIB-D). ⁵ The result also supports the consistent subtype definition in our data.

7. Inconsistent figure quality is a major concern and makes interrogating figures very difficult in numerous instances.

Response: Thanks for the reviewer's advice. We had revised most of our figures, including Figure 1-2, 5-7, and all supplementary figures. The font size of these figures had been enlarged, and the order of sub-plots has been changed. Also, we had added more figures to support our conclusion in our manuscript as we had described above.

Minor Concerns:

1. Figure II B is very difficult to interpret- the figures appear to be processed differently and maybe have different pixel size.

Response: Much thanks for the reviewer's advice. It would be confusing to interpret the conclusion that IGF1 is upregulated and CDKN1A is downregulated in PA group. Hence, we had added the violin plots to directly show the difference of *IGF1* and *CDKN1A* expression between PA and AC group. We observed the significantly higher expression of *IGF1* (0.01 vs. -0.04) and lower expression of *CDKN1A* (-0.01 vs. 0.05) in PA group. And we also revised the corresponding part in our manuscript.

2. A number of supplemental figures would be useful in the main figures since you can't fully understand the presented figures without the additional data. For example the

cluster diversity maps showing the different cluster regions- this is key data to understand how genes are expressed throughout the tissue.

Response: Much thanks for the reviewer's advice. In our revised version manuscript. We had added several figures to support our conclusions. We had compared the top marker genes in our clusters to those in other publications (Figure IIB/III/VIB-D). Furthermore, singleR analysis was used to validate the population identities of cluster 3, 4, 15 (Figure IIA). Also, the violin plots of *IGF1* and *CDKN1A* were used for showing the difference of expression in PA and AC group (Figure IVF). Lastly, we had validated the key cell culture experiments with **bone marrow derived macrophages (BMDMs)** (Figure IX) and the inverse correlations of IL-1 β and STAT3 through *in vivo* experiments by loss-of-function of IL-1 β (Figure 8). And we hope these supplementary figures could help our manuscript more clarity.

3. Plots in Figure 1 are low resolution and pixelated when zooming in- especially the violin. Please provide higher resolution images to make it easier to examine cluster variation.

Response: Much thanks for the reviewer's advice. We had revised Figure 1 to get higher resolution image. Also, the violin plot has been re-plotted. We added the datapoints and changed the color of the violin. And we hope it could be easier to read and catch the key point.

4. What is being concluded by the monocle analysis? It may be more informative to examine just smooth muscle derived cells.

Response: The cells in cluster 3, 4, and 15, which are defined as VSMCs, are included in the following Monocle analysis. To make it clarity, we had added this sentence at the first paragraph. And, we felt so sorry to examine just smooth muscle derived cells. However, it almost impossible conduct the pseudotime analysis in human smooth muscle derived cells as the experimental and ethic restrictions. Because we could not obtain the atherosclerotic tissues from patients.

5. A series of typos or writing mistakes- Lines 123, 134, 225, 329, etc.

Response: the sentence in 123 had been revised as "In turn, macrophage regulates the clonality and apoptosis of VSMCs during atherosclerosis". The sentence in 14 had been revised as "Single-cell transcriptomic data (10x Genomics) of AC and patient-matched PA portions of plaques from 3 patients undergoing carotid endarterectomy (GSE159677) were obtained from the Gene Expression Omnibus (GEO) database". The sentence in line 225 had been revised as "Cells and tissue were lysed in lysis buffer for 60 min, followed by centrifugation at 12,000 \times g for 20 min". The sentence in line 329 had been revised as "We then analyzed activity changes for those 52 TFs among different cell states".

6. What is being compared in Figure 3G?

Response: Much thanks for the reviewer's advice. It could be a little confusing without adequate description. Firstly, we conducted BEAM analysis in order to identify the different genes associated with the second branch node. And the genes were assigned into 3 different clusters according to the gene expression pattern in VSMCs' different states (Figure 3E). We found that genes cluster 1 is upregulated in state 4 and gene cluster 2 is highly expressed in state 3. Then the KEGG analysis had been performed to compare the top pathways between state 3 and state 4 (Figure 3G). And to make it clarity, we had added one sentence to explain how the gene cluster has been defined. And the sentence has been written as "And then, the

genes with branch-dependent expression were assigned them to 3 gene clusters according to trajectory differentiation to identify the mechanism by which the VSMC fate decision is made”.

7. Figure legends are minimal and could be expanded to assist the reader in understanding how data is being presented.

Response: Much thanks for the reviewer’s advice. The figure and supplementary figure legends have been expanded according to the reviewer’s advice. We had added the detailed description of the statistic method, number per group, symbol instructions. And we hope the newly-submitted figure legends could reach the reviewer’s requirements.

Reference

1. Wirka RC, *et al.* Atheroprotective roles of smooth muscle cell phenotypic modulation and the TCF21 disease gene as revealed by single-cell analysis. *Nature medicine* **25**, 1280-1289 (2019).
2. Rao M, *et al.* Resolving the intertwining of inflammation and fibrosis in human heart failure at single-cell level. *Basic research in cardiology* **116**, 55 (2021).
3. Depuydt MAC, *et al.* Microanatomy of the Human Atherosclerotic Plaque by Single-Cell Transcriptomics. *Circulation research* **127**, 1437-1455 (2020).
4. Fernandez DM, *et al.* Single-cell immune landscape of human atherosclerotic plaques. *Nature medicine* **25**, 1576-1588 (2019).
5. Zernecke A, *et al.* Meta-Analysis of Leukocyte Diversity in Atherosclerotic Mouse Aortas. *Circulation research* **127**, 402-426 (2020).
6. Grootaert MOJ, Bennett MR. Vascular smooth muscle cells in atherosclerosis: Time for a reassessment. *Cardiovascular research*, (2021).
7. Stuart T, *et al.* Comprehensive Integration of Single-Cell Data. *Cell* **177**, 1888-1902.e1821 (2019).
8. Willemsen L, de Winther MP. Macrophage subsets in atherosclerosis as defined by single-cell technologies. *The Journal of pathology* **250**, 705-714 (2020).
9. Zhang F, Guo X, Xia Y, Mao L. An update on the phenotypic switching of vascular smooth muscle cells in the pathogenesis of atherosclerosis. *Cellular and molecular life sciences : CMLS* **79**, 6 (2021).

10. Yap C, Mieremet A, de Vries CJM, Micha D, de Waard V. Six Shades of Vascular Smooth Muscle Cells Illuminated by KLF4 (Kruppel-Like Factor 4). *Arteriosclerosis, thrombosis, and vascular biology* **41**, 2693-2707 (2021).
11. Sorokin V, *et al.* Role of Vascular Smooth Muscle Cell Plasticity and Interactions in Vessel Wall Inflammation. *Frontiers in immunology* **11**, 599415 (2020).
12. Vengrenyuk Y, *et al.* Cholesterol loading reprograms the microRNA-143/145-myocardin axis to convert aortic smooth muscle cells to a dysfunctional macrophage-like phenotype. *Arteriosclerosis, thrombosis, and vascular biology* **35**, 535-546 (2015).
13. Feil S, *et al.* Transdifferentiation of vascular smooth muscle cells to macrophage-like cells during atherogenesis. *Circulation research* **115**, 662-667 (2014).
14. Wu JH, *et al.* Drebrin attenuates atherosclerosis by limiting smooth muscle cell transdifferentiation. *Cardiovascular research* **118**, 772-784 (2022).
15. Bao Z, *et al.* Advanced Glycation End Products Induce Vascular Smooth Muscle Cell-Derived Foam Cell Formation and Transdifferentiate to a Macrophage-Like State. *Mediators of inflammation* **2020**, 6850187 (2020).
16. Alencar GF, *et al.* Stem Cell Pluripotency Genes Klf4 and Oct4 Regulate Complex SMC Phenotypic Changes Critical in Late-Stage Atherosclerotic Lesion Pathogenesis. *Circulation* **142**, 2045-2059 (2020).
17. Liu M, Gomez D. Smooth Muscle Cell Phenotypic Diversity. *Arteriosclerosis, thrombosis, and vascular biology* **39**, 1715-1723 (2019).
18. Xiong D, Wang Y, You M. A gene expression signature of TREM2(hi) macrophages and $\gamma\delta$ T cells predicts immunotherapy response. *Nature communications* **11**, 5084 (2020).

Reviewers' comments:

Reviewer #1 (Remarks to the Author):

the major claim of the paper is that macrophages signal to vSMCs and this alters their pro-inflammatory nature/function. specifically IL1b by macrophages would limit stat3B
the authors first describe transcriptome of myeloid and vSMCS, predict their interaction, and then study the predicted interaction in vitro and in vivo.

The topic of vSMCs and the regulation thereof is of great interest to the field, and other have studied vSMC intrinsic regulation (Wirka, Pan et al, Alencar et al, Wang CVR 2022 (10.1093/cvr/cvab347), and less so by macrophages, but some evidence exists (<https://pubmed.ncbi.nlm.nih.gov/35381443/>)
Changes really increased strength of the biological message, but this is not sufficient to support the major claim.

w/o SMC or macrophage reporters or myeloid specific IL1b knockout the authors do not conclusively show that IL1b derived from macrophages affects STAT3B in SMCs. In vitro no functional changes are shown in SMC function, merely changes in marker expression. These experiments are need to influence the field.

Minor:

although I appreciate the new sgRNA study, the authors do not describe its control. what was negative control; vehicle or empty vector AAV? proper control would be one of an empty vector or scrambled gRNA.

please confirm downregulation of Il1beta in figure 8. As the plaque and macrophage content is virtually gone, as per images, conclusion on regulation is limited: it can not be solely due to cell-communication blocking effects, but rather a loss of cd68+ cells. also the immunofluorescence does not support the source of Il1beta being CD58+ macrophages or cd68+ VSMCs

Reviewer #2 (Remarks to the Author):

In the revised manuscript the authors provide additional experimental data and bioinformatics analysis to support the narrative that plaque associated macrophage promote smooth muscle differentiation toward a CD68+ lineage during atherosclerosis progression. Overall, the manuscript is greatly improved and story is of interest to the field. However, I still have a few minor concerns that I believe should be addressed by the authors.

(Line references are using the marked version of the manuscript)

Acknowledgements section: The sentence "And we also give Mr. Torkamani and his group..." needs to be edited to be more professional. Something more similar to other acknowledgements they write in the section, such as "We thank Dr. Ali Toramani (Scripps Research Institute) and his group for generating and publicly sharing the single cell dataset utilized in this study".

In addition, it is imperative that the authors formally cite the original BioRxiv paper in the methods section ~line 133 when they reference the scRNA-seq dataset, (Alsaigh, BioRxiv 2020). And also cite the original dataset for the RNA-seq data ~line 193 (Ayari, J. Biosci 2013).

Line 353: Figure 5D and 5E are referenced in the text, "Cells of cluster 0 were recognized as TREM2^{high} owing to their high expressing TREM2, CD9, LGALS3, CTSB, and ABCA1 (Figure III VD-E and III VG)." Cluster 0 has very little Trem2 expression. I think it would be appropriate to say that the gene signature of this cluster is consistent with the Trem2⁺ macrophages previously defined. It is probably misleading to say that they are "high expressing" Trem2 as it is currently written in the text, especially considering that Figure 5E shows that Trem2 is primarily enriched in Cluster 1.

Line 355: CX3CR1 is not an embryonic marker and is highly expressed on monocytes and often monocyte-derived macrophages. The citations for this claim are made out of context. It would be more appropriate to say that CX3CR1 is enriched in Cluster 1. To claim it has an association with cellular origin would require significant additional work.

Line 560: "However, Gary et. Al., reported..." this sentence is not written correctly nor is it a thoughtful interpretation from the published paper. The paper should be cited as "Gomez et. al". and, it would be helpful if data from the paper is specifically discussed, rather than a broad and non-specific claim regarding the priority of IL-1 function in atherosclerotic lesions.

The Figures are improved quality, but some remain difficult to view. An example is Figure 4. Even as a full page figure, the panels are nearly impossible to read.

There are still some typos, I caught a few below:

Line 178: "Nichnet" should be Nichenet.

Line 313: "Wrika" should be Wirka

Line 458: "Activate" should be Active

Line 541: "Validate" should be validated

Response to Reviewer #1

the major claim of the paper is that macrophages signal to vSMCs and this alters their pro-inflammatory nature/function. specifically IL1b by macrophages would limit stat3B.

the authors first describe transcriptome of myeloid and vSMCS, predict their interaction, and then study the predicted interaction in vitro and in vivo.

The topic of vSMCs and the regulation thereof is of great interest to the field, and other have studied vSMC intrinsic regulation (Wirka, Pan et al, Alencar et al, Wang CVR 2022 (10.1093/cvr/cvab347), and less so by macrophages, but some evidence exists (<https://pubmed.ncbi.nlm.nih.gov/35381443/>)

Changes really increased strength of the biological message, but this is not sufficient to support the major claim.

Response: Much thanks for the reviewer's valuable comments. According to your suggestions, we had added the functional experiments of VSMC, additional descriptions for AAV-NC, and validation of IL-1 β expression in aorta. As it's hard to conduct VSMC/macrophage reporter or specific myeloid IL1B knockout experiments due to restriction of our laboratory condition. We had revised the corresponding claims with a general tone-down in our conclusion. We wish the newly-submitted manuscript would meet your requirements.

w/o SMC or macrophage reporters or myeloid specific IL1b knockout the authors do not conclusively show that IL1b derived from macrophages affects STAT3B in SMCs. In vitro no functional changes are shown in SMC function, merely changes in marker expression. These experiments are need to influence the field.

Response: Much thanks for the reviewer's advice. In our study, we could find the macrophage-VSMC crosstalk could induce the trans-differentiation of VSMC and functional alterations in macrophage-like VSMC. And this crosstalk is at least partially determined by macrophage-derived IL-1 β based on our *in vitro* experiments. However, myeloid specific IL-1 β knockout by CRISPR-Cas9 in cells or Cre-loxP technology in animals or VSMC/macrophage reporter experiments could provide fully conclusive evidences in the effect of macrophage-derived IL-1 β on STAT3 in macrophage-like VSMCs as the reviewer's comment. To be honest, it's difficult to conduct these experiments based on our laboratory condition and technology. And according to the editor's requirement, we had modified our sentences with general tone-down in our conclusions which pertaining to IL-1 β -STAT3 axis involved in macrophage-macrophage-like VSMC crosstalk, and declared this limitation in our discussion part. The sentence at line 54 was modified as "The current findings provide new insight into the crosstalk between macrophages and macrophage-like VSMCs, which **would** drive functional alterations in the latter cell type during atherosclerosis progression." The sentence at line 510 was revised as "Furthermore, IL-1 β **may** secreted by macrophage induced macrophage-like VSMC functional alterations through the inhibition of *STAT3*, which was confirmed through *in vitro* analyses". The sentence at line 550 was modified as "Next, we validated the **potential** macrophage-to-macrophage-like VSMC crosstalk via IL-1 β /STAT3 axis through *in vitro* experiments". Accordingly, we hope the revised conclusion would be more appropriate to discuss the IL-1 β /STAT3 axis in macrophage-

macrophage-like VSMC crosstalk. Also, we had added one sentence in limitation part (line 593) as “Thirdly, myeloid specific IL-1 β knockout by CRISPR-Cas9 in cells or Cre-loxP technology in animals or VSMC/macrophage reporter experiments could provide conclusive evidences in the effect of macrophage-derived IL-1 β on STAT3 in macrophage-like VSMCs. But it’s unable to conduct these experiments in our study due to restrictions of laboratory condition”.

Furthermore, we had conducted Oil-red O staining and cell migration assays to explore the functional alterations in macrophage-like VSMCs with RM incubation and addition of IL-1Ra (Figure XA). The Oil-red O staining results demonstrated the intracellular lipid droplets were significantly increased at 48h and decreased in response to IL-1Ra treatment (Figure XB). These findings indicate that IL-1 β contributes the lipid accumulation in macrophage-like VSMCs. Furthermore, we also conducted wound healing tests to identify the effect of IL-1 β on macrophage-like VSMC migration. Consistent with declined lipid accumulation, IL-1Ra also abrogated the promotive effect of RM treatment on migration (Figure XC). Also, these methods and results have been added into our method and result parts (line 271-280, 482-487). We hope these revisions could address your concerns about functional changes in VSMCs.

Minor:

although I appreciate the new sgRNA study, the authors do not describe its control. what was negative control; vehicle or empty vector AAV? proper control would be one of an empty vector or scrambled sgRNA.

Response: Much thanks for the reviewer’s advice. The negative control for AAV-sgIL-1 β in our study is the scramble sgRNA. To make it clear in our manuscript, the sentence at line 217 were revised as “To inhibit the expression of IL-1 β , the mice were fixed to a stereo-locator and injected through the tail vein with 50 μ l of adeno-associated virus-IL-1 β (AAV-sgIL-1 β) (CMV-NLS-SaCas9-NLS-3xHA-bGHpA-U6-sgRNA, Genechem Co., Ltd., Shanghai, China) and negative control (scramble sgRNA, AAV-NC) at 8th week of HFD.” Furthermore, we also modified the group names as “AS 16w AAV-NC” and “AS 16w AAV-sgIL-1 β ” in Figure 8 and Figure XI. Also, we revised our result to keep the consistency of group names in animal experiments.

please confirm downregulation of Il1beta in figure 8. As the plaque and macrophage content is virtually gone, as per images, conclusion on regulation is limited: it cannot be solely due to cell-communication blocking effects, but rather a loss of cd68+ cells. also the immunofluorescence does not support the source of Il1beta being CD68+ macrophages or cd68+ VSMCs

Response: Much thanks for the reviewer’s suggestion. We had added the western blot results in Figure XI for validation. We observed IL-1 β expression is increased in aorta of ApoE^{-/-} mice fed with HFD for 16 weeks compared with those fed for 8 weeks. Furthermore, the increase of IL-1 β can be reversed by injection with AAV-sgIL-1 β (Figure XIB).

The conclusion from immunofluorescence assay is limited as it cannot provide lineage-tracing evidences in macrophage-VSMC communications. Hence, the sentence in result part to describe the immunofluorescence (line 495) was revised as “Using

immunofluorescence, expression of p-STAT3 was also repressed by IL-1 β in **CD68+ α SMA+ cells**, which were shown to accumulate in the plaque during atherosclerosis progression (Figure 8E-G)” to make the description more accurate. Furthermore, as the macrophage-VSMC crosstalk cannot be validated in our immunofluorescence assays, we could only find the upregulation of p-STAT3 in CD68+ α SMA+ cells after IL-1 β inhibition. Hence the regulation conclusion at line **580** was revised as “If such macrophage-to-VSMC interactions within atherosclerotic plaque are not blocked via anti-inflammatory therapeutics, such as canakinumab, macrophage-like VSMCs undergo enhanced adhesion, inflammation, and apoptosis thus **potentially** contributing to atherosclerosis progression.”

Response to Reviewer #2

In the revised manuscript the authors provide additional experimental data and bioinformatics analysis to support the narrative that plaque associated macrophage promote smooth muscle differentiation toward a CD68+ lineage during atherosclerosis progression. Overall, the manuscript is greatly improved and story is of interest to the field. However, I still have a few minor concerns that I believe should be addressed by the authors.

(Line references are using the marked version of the manuscript)

Response: Much thanks for the reviewer's valuable comments. We had revised the font size of our figures, citations, acknowledgement, typos, inaccurate descriptions, and discussion about Gomez's study. And we wish the newly submitted manuscript could meet your requirements for acceptance.

Acknowledgements section: The sentence "And we also give Mr. Torkamani and his group..." needs to be edited to be more professional. Something more similar to other acknowledgements they write in the section, such as "We thank Dr. Ali Toramani (Scripps Research Institute) and his group for generating and publicly sharing the single cell dataset utilized in this study".

Response: Much thanks for the reviewer's suggestion. We had revised this sentence (line 613) as "We thank Dr. Ali Toramani (Scripps Research Institute) and his group for generating and publicly sharing the single cell dataset utilized in this study" according to the reviewer's comment.

In addition, it is imperative that the authors formally cite the original BioRxiv paper in the methods section ~line 133 when they reference the scRNA-seq dataset, (Alsaigh, BioRxiv 2020). And also cite the original dataset for the RNA-seq data ~line 193 (Ayari, J. Biosci 2013).

Response: Much thanks for the reviewer's advice. We had cited the corresponding articles of these two public datasets to respect for their selfless contributions.

Line 353: Figure 5D and 5E are referenced in the text, "Cells of cluster 0 were recognized as TREM2^{high} owing to their high expressing TREM2, CD9, LGALS3, CTSB, and ABCA1 (Figure III VD-E and III VG)." Cluster 0 has very little Trem2 expression. I think it would be appropriate to say that the gene signature of this cluster is consistent with the Trem2⁺ macrophages previously defined. It is probably misleading to say that they are "high expressing" Trem2 as it is currently written in the text, especially considering that Figure 5E shows that Trem2 is primarily enriched in Cluster 1.

Response: Much thanks for the reviewer's advice. We had revised this sentence (line 348-350) as "Cells of cluster 0 were recognized as *TREM2^{high}* owing to their high expressing *CD9*, *LGALS3*, *CTSB*, and *ABCA1*, which are gene markers of Trem2⁺ macrophage as previously defined (Figure VD-E and VG)". And we hope the modified sentence is more accurate to address why cluster 0 has been defined as *TREM2^{high}* subtype.

Line 355: CX3CR1 is not an embryonic marker and is highly expressed on monocytes and often monocyte-derived macrophages. The citations for this claim

are made out of context. It would be more appropriate to say that CX3CR1 is enriched in Cluster 1. To claim it has an association with cellular origin would require significant additional work.

Response: Much thanks for the reviewer's advice. At steady state, vascular macrophages are mainly derived from CX3CR1+ embryonic precursor cells with little contribution from blood monocytes.¹ But, CX3CR1 is not the embryonic marker and presents on different leukocyte subsets.² In atherogenesis process, high expression of CX3CR1 in T cell and monocyte/macrophage play a critical role in plaque.^{3, 4, 5} Accordingly, we had deleted "embryonic marker" at line 350 to avoid misunderstanding.

Line 560: "However, Gary et. Al., reported..." this sentence is not written correctly nor is it a thoughtful interpretation from the published paper. The paper should be cited as "Gomez et. al". and, it would be helpful if data from the paper is specifically discussed, rather than a broad and non-specific claim regarding the priority of IL-1 function in atherosclerotic lesions.

Response: Much thanks for the reviewer's advice. In this paragraph, we want to discuss the controversial role of IL-1 β in atherosclerosis. Hence, we provided the main conclusion of different significant studies in this field without deeply discussion. The sentence at line 562 about Gomez's study is not written appropriate in our discussion. Hence, this sentence had been revised as "IL-1 β may be the double-edged sword in atherosclerosis. Gomez and colleagues reported that IL-1 β antibody treatment to Apoe^{-/-} mice between 18 and 26 weeks of Western diet feeding induced a marked reduction in VSMC and collagen content and accompanied by increased lesional macrophages. Specifically, VSMC-selective *Il1r1* KO resulted in smaller lesions nearly devoid of SMC and a fibrous cap". And we also revised the sentence at line 563 as "Our study also supported the detrimental role of IL-1 β in atherosclerotic progression at a single-cell resolution and provided evidence for the potential mechanisms underlying the atherogenic effect of IL-1 β on VSMCS" for more accurate description.

And we thought the potential reasons underlying the seemingly contradictory conclusion from Gomez's and CANTOS studies are as follows. Firstly, although reduced collagen content and increased LGALS3+ cells exhibited in fibrous cap, intraplaque hemorrhage detected by Ter119 staining between *Il1r1*^{SMC Δ/Δ} and *Il1r1*^{SMC WT/WT} mice groups showed no difference in Figure 5e-f. Accordingly, there is no direct clinical evidences/events to prove the plaque instability in their animal experiments. Furthermore, a novel dual lineage tracing model indicated that VSMCs transition through Lgals3 activation state take on multiple phenotypes.⁶ The conclusion probably suggested the increased LGALS3+ cells in *Il1r1*^{SMC Δ/Δ} mice groups may not only derived from macrophages. Moreover, both two CANTOS studies did not provide the immunohistochemistry or plaque specimens data from patients treated with canakinumab.^{7, 8} So more evidences should be conducted in future experiments to explore the gap between detrimental role of IL-1 β in human plaque and potential atheroprotective effect on late stage of atherosclerosis.

The Figures are improved quality, but some remain difficult to view. An example is Figure 4. Even as a full page figure, the panels are nearly impossible to read.

Response: Much thanks for the reviewer's advice. We had also observed the inconsistent figure quality in our manuscript, especially small labels to view. We had enlarged the font size in Figure 1E, Figure 2D, Figure 2F, the whole Figure 4, Figure II, Figure III, and Figure V. The labels of subplots in Figure IX had also been rearrangement to avoid misleading. We had also provided the support table (Table 11) for Figure 4A as the row annotation labels is not enough large due layout restriction. We wish the newly revised figures would be easier to read.

There are still some typos, I caught a few below:

Line 178: "Nichnet" should be Nichenet.

Response: We had corrected this typo according to the reviewer's suggestion.

Line 313: "Wrika" should be Wirka

Response: We had replaced "Wrika's" as "Wirka's" according to the reviewer's advice.

Line 458: "Activate" should be Active

Response: We had revised "activate" as "active" according to the reviewer's comment.

Line 541: "Validate" should be validated

Response: We had modified "validate" as "validated" according to the reviewer's advice.

Reference

1. Ensan S, *et al.* Self-renewing resident arterial macrophages arise from embryonic CX3CR1(+) precursors and circulating monocytes immediately after birth. *Nature immunology* **17**, 159-168 (2016).
2. Imai T, *et al.* Identification and molecular characterization of fractalkine receptor CX3CR1, which mediates both leukocyte migration and adhesion. *Cell* **91**, 521-530 (1997).
3. Quintar A, *et al.* Endothelial Protective Monocyte Patrolling in Large Arteries Intensified by Western Diet and Atherosclerosis. *Circulation research* **120**, 1789-1799 (2017).
4. Bonacina F, *et al.* Adoptive transfer of CX3CR1 transduced-T regulatory cells improves homing to the atherosclerotic plaques and dampens atherosclerosis progression. *Cardiovascular research* **117**, 2069-2082 (2021).
5. McArdle S, *et al.* Migratory and Dancing Macrophage Subsets in Atherosclerotic Lesions. *Circulation research* **125**, 1038-1051 (2019).
6. Alencar GF, *et al.* Stem Cell Pluripotency Genes Klf4 and Oct4 Regulate Complex SMC Phenotypic Changes Critical in Late-Stage Atherosclerotic Lesion Pathogenesis. *Circulation* **142**, 2045-2059 (2020).
7. Everett BM, MacFadyen JG, Thuren T, Libby P, Glynn RJ, Ridker PM. Inhibition of Interleukin-1 β and Reduction in Atherothrombotic Cardiovascular

Events in the CANTOS Trial. *Journal of the American College of Cardiology* **76**, 1660-1670 (2020).

8. Ridker PM, *et al.* Antiinflammatory Therapy with Canakinumab for Atherosclerotic Disease. *The New England journal of medicine* **377**, 1119-1131 (2017).

REVIEWERS' COMMENTS:

Reviewer #1 (Remarks to the Author):

The changes in wording, and extra functional experiments, and controls strengthen the manuscript. One minor comment remains regarding the AAV. The type of AAV was not mentioned, and this determines its tropism. Generally, none target the aorta well/exclusively, rather the liver is targeted (AAV8). This results in circulating levels of gRNA, excluding a vascular-specific effect in line with a systemic treatment. So while not specific for aorta macrophages, it possibly reflects the effect of canakinumab. Please append methods and discussion

Reviewer #2 (Remarks to the Author):

Thank you for addressing my concerns.

Reviewer #1 (Remarks to the Author):

The changes in wording, and extra functional experiments, and controls strengthen the manuscript.

One minor comment remains regarding the AAV. The type of AAV was not mentioned, and this determines its tropism. Generally, none target the aorta well/exclusively, rather the liver is targeted (AAV8). This results in circulating levels of gRNA, excluding a vascular-specific effect in line with a systemic treatment. So while not specific for aorta macrophages, it possibly reflects the effect of canakinumab. Please append methods and discussion

Response: Much thanks for the reviewer's useful advice. AAV9 was chosen to inhibit the expression of IL-1 β in our study for its rapid onset, high expression, and best viral genome distribution among AAV1-9.¹ And, several studies adopted AAV9 to regulate the gene expression in atherosclerosis.^{2,3,4} Furthermore, we supposed that the effect of AAV9 therapy (AAV-sgIL-1 β) is similar to canakinumab as the reviewer's opinion. Also, we had added the description of AAV-sg-IL1 β in method part (line 551). Also, we had discussed the strengths and similarity of AAV9 with canakinumab in our discussion part (line 428-432). And we hope these revisions could meet your requirements.

Reference

1. Zincarelli C, Soltys S, Rengo G, Rabinowitz JE. Analysis of AAV serotypes 1-9 mediated gene expression and tropism in mice after systemic injection. *Molecular therapy : the journal of the American Society of Gene Therapy* **16**, 1073-1080 (2008).
2. Kumar S, Kang DW, Rezvan A, Jo H. Accelerated atherosclerosis development in C57Bl6 mice by overexpressing AAV-mediated PCSK9 and partial carotid ligation. *Laboratory investigation; a journal of technical methods and pathology* **97**, 935-945 (2017).
3. Zhao Y, *et al.* Deacetylation of Caveolin-1 by Sirt6 induces autophagy and retards high glucose-stimulated LDL transcytosis and atherosclerosis formation. *Metabolism: clinical and experimental* **131**, 155162 (2022).
4. Chen Q, *et al.* Recombinant adeno-associated virus serotype 9 in a mouse model of atherosclerosis: Determination of the optimal expression time in vivo. *Molecular medicine reports* **15**, 2090-2096 (2017).